# Collapsed Inference for Bayesian Deep Learning

**Zhe Zeng**
Computer Science Department
University of California, Los Angeles
zhezeng@cs.ucla.edu

**Guy Van den Broeck**
Computer Science Department
University of California, Los Angeles
guyvdb@cs.ucla.edu

## Abstract

Bayesian neural networks (BNNs) provide a formalism to quantify and calibrate uncertainty in deep learning. Current inference approaches for BNNs often resort to few-sample estimation for scalability, which can harm predictive performance, while its alternatives tend to be computationally prohibitively expensive. We tackle this challenge by revealing a previously unseen connection between inference on BNNs and *volume computation* problems. With this observation, we introduce a novel collapsed inference scheme that performs Bayesian model averaging using *collapsed samples*. It improves over a Monte-Carlo sample by limiting sampling to a subset of the network weights while pairing it with some closed-form conditional distribution over the rest. A collapsed sample represents uncountably many models drawn from the approximate posterior and thus yields higher sample efficiency. Further, we show that the marginalization of a collapsed sample can be solved analytically and efficiently despite the non-linearity of neural networks by leveraging existing volume computation solvers. Our proposed use of collapsed samples achieves a balance between scalability and accuracy. On various regression and classification tasks, our collapsed Bayesian deep learning approach demonstrates significant improvements over existing methods and sets a new state of the art in terms of uncertainty estimation as well as predictive performance.

## 1 Introduction

Uncertainty estimation is crucial for decision making. Deep learning models, including those in safety-critical domains, tend to estimate uncertainty poorly. To overcome this issue, Bayesian deep learning obtains a posterior distribution over the model parameters hoping to improve predictions and provide reliable uncertainty estimates. Among Bayesian inference procedures with neural networks, Bayesian model averaging (BMA) is particularly compelling (Wasserman, 2000; Fragoso et al., 2018; Maddox et al., 2019). However, computing BMAs is distinctly challenging since it involves marginalizing over posterior parameters, which possess some unusual topological properties such as mode-connectivity (Izmailov et al., 2021). We show that even with simple low-dimensional approximate parameter posteriors as uniform distributions, doing BMA requires integrating over highly *non-convex* and *multi-modal* distributions with discontinuities arising from non-linear activations (cf. Figure 1a). Accurately approximating the BMA can achieve significant performance gains (Izmailov et al., 2021). Existing methods mainly focus on general-purpose MCMC, which can fail to converge, or provides inaccurate few-sample predictions (Kristiadi et al., 2022), because running longer sampling chains is computationally expensive, and variational approaches that typically use a mean-field approximation that ignores correlations induced by activations (Jospin et al., 2022).

In this work, we are interested in developing *collapsed samplers*, also known as *cutset* or *Rao-Blackwellised* samplers for BMA. A collapsed sampler improves over classical particle-based methods by limiting sampling to a subset of variables and further pairing each sample with a closed-form

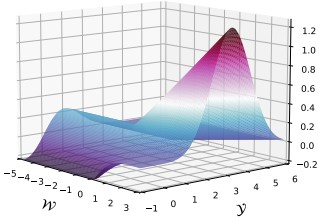 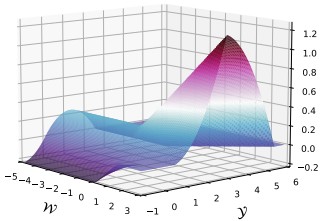

(a) $p(y \mid \boldsymbol{x}, \boldsymbol{w})$ being Gaussian. See Example 3.

(b) $p(y \mid \boldsymbol{x}, \boldsymbol{w})$ being triangular. See Section 3.

Figure 1: The integral surface of (a) the expected prediction in BMA, and (b) our proposed approximation. Both are highly non-convex and multi-modal. The z-axis is the weighted prediction $y\, p(y \mid \boldsymbol{x}, \boldsymbol{w})\, p(\boldsymbol{w} \mid \mathcal{D})$. Integration of (a) does not admit a closed-form solution, yet integration of (b) is a close approximation that can be solved exactly and efficiently by WMI solvers.

representation of a conditional distribution over the rest whose marginalization is often tractable. Collapsed samplers are effective at variance reduction in graphical models (Koller & Friedman, 2009), however no collapsed samplers are known for Bayesian deep learning. We believe that this is due to the lack of a closed-form marginalization technique congruous with the non-linearity in deep neural networks. Our aim is to overcome this issue and improve BMA estimation by incorporating exact marginalization over (close approximate) conditional distributions into the inference scheme. Nevertheless, scalability and efficiency are guaranteed by the sampling part of our proposed algorithm.

Marginalization is made possible by our observation that BMA reduces to weighted volume computation. Certain classes of such problems can be solved exactly by so-called weighted model integration (WMI) solvers (Belle et al., 2015a). By closely approximating BMA with WMI, these solvers can provide accurate approximations to marginalization in BMA (cf. Figure 1b). With this observation, we propose CIBER, a collapsed sampler that uses WMI for computing conditional distributions. In the few-sample setting, CIBER delivers more accurate uncertainty estimates than the gold-standard Hamiltonian Monte Carlo (HMC) method (cf. Figure 2). We further evaluate the effectiveness of CIBER on regression and classification benchmarks and show significant improvements over other Bayesian deep learning approaches in terms of both uncertainty estimation and accuracy.

## 2 Bayesian Model Averaging as Weighted Volume Computation

In **Bayesian Neural Networks (BNN)**, given a neural network $f_{\boldsymbol{w}}$ parameterized by weights $\boldsymbol{w}$, instead of doing inference with deterministic $\boldsymbol{w}$ that optimize objectives such as cross-entropy or mean squared error, Bayesian learning infers a posterior $p(\boldsymbol{w} \mid \mathcal{D})$ over parameters $\boldsymbol{w}$ after observing data $\mathcal{D}$. During inference, this posterior distribution is then marginalized to produce final predictions. This process is called **Bayesian Model Averaging (BMA)**. It can be seen as learning an ensemble of an infinite number of neural nets and aggregating their results. Formally, given input $\boldsymbol{x}$, the posterior predictive distribution and the expected prediction for a regression task are

$$p(y \mid \boldsymbol{x}) = \int p(y \mid \boldsymbol{x}, \boldsymbol{w})\, p(\boldsymbol{w} \mid \mathcal{D})\, d\boldsymbol{w}, \qquad \text{and} \qquad \mathbb{E}_{p(y\mid\boldsymbol{x})}[y] = \int y\, p(y \mid \boldsymbol{x})\, dy. \quad (1)$$

For classification, the (most likely) prediction is the class $\arg\max_y p(y \mid \boldsymbol{x})$. BMA is intuitively attractive because it can be risky to base inference on a single neural network model. The marginalization in BMA gets around this issue by averaging over models according to a Bayesian posterior.

BMA requires approximations to compute posterior predictive distributions and expected predictions, as the integrals in Equation 1 are intractable in general. Deriving efficient and accurate approximations remains an active research topic (Izmailov et al., 2021). We approach this problem by observing that the marginalization in BMA with ReLU neural networks can be cast as weighted volume computation (WVC). Later we show that it can be generalized to any neural network when combined with sampling. In WVC, various tools exist for solving certain WVC problem classes (Baldoni et al.,

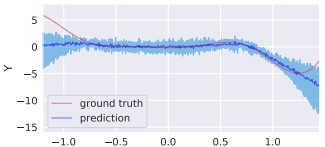 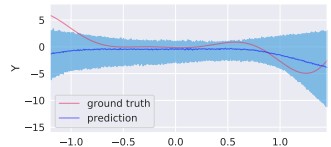 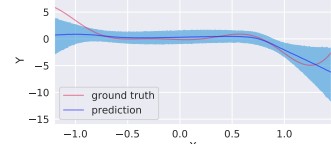

(a) HMC with 10 samples   (b) CIBER with 10 collapsed samples   (c) HMC with $2k$ samples

Figure 2: Uncertainty estimates for regression. The red line is the ground truth. The dark blue line shows the predictive mean. The shaded region is the $90\%$ confidence interval of the predictive distribution. For the same number of samples, (b) CIBER is closer than (a) small-sample HMC to (c) a highly accurate but slow HMC with a large number of samples. See the Appendix for details.

2014; Kolb et al., 2019; Zeng et al., 2020c). This section reveals the connection between BMA and WVC. It opens up a new perspective for developing BMA approximations by leveraging WVC tools.

**Definition 1** (WVC). *A weighted volume computation (WVC) problem is defined by a pair $(\boxempty, \phi)$ where a region $\boxempty$ is a conjunction of arithmetic constraints and weight $\phi : \boxempty \to \mathbb{R}$ is an integrable function assigning weights to elements in $\boxempty$. The task of WVC is to compute the integral $\int_{\boxempty} \phi(\boldsymbol{x}) \, d\boldsymbol{x}$.*

## 2.1 A Warm-Up Example

Consider a simple yet relevant setting where the predictive distribution $p(y \mid \boldsymbol{x}, \boldsymbol{w})$ is a Dirac delta distribution with zero mass everywhere except at $f_{\boldsymbol{w}}(\boldsymbol{x})$, such that $\int y \, p(y \mid \boldsymbol{x}, \boldsymbol{w}) \, dy = f_{\boldsymbol{w}}(\boldsymbol{x})$.

**Example 2.** *Assume a model $f_{\boldsymbol{w}}(x) = \mathsf{ReLU}(w \cdot x)$ with a uniform posterior over the parameter: $p(w \mid \mathcal{D}) = \frac{1}{6}$ with $w \in [-3, 3]$. Let the input be $x = 1$. For parameter $w \in [-3, 0]$, the model $f_{\boldsymbol{w}}$ always predicts $0$, and otherwise (i.e., $w \in (0, 3]$), it predicts $w$. Thus, the expected prediction (Equation 1) is $\mathbb{E}_{p(y|\boldsymbol{x})}[y] = \int_{\boxempty_\perp} 0 \cdot \frac{1}{6} \, d\boldsymbol{w} + \int_{\boxempty_\top} w \cdot \frac{1}{6} \, d\boldsymbol{w}$. That is, a summation of two WVC problems $(\boxempty_\perp, 0)$ and $(\boxempty_\top, w/6)$ with $\boxempty_\perp = (-3 \le w \le 0)$ and $\boxempty_\top = (0 \le w \le 3)$. The BMA integral decomposes into WVC problems with different weights due to the $\mathsf{ReLU}$ activation.*

These WVC problems have easy closed-form solutions. This is no longer the case in the following.

**Example 3.** *Assume a model $f_{\boldsymbol{w}}(x)$ and posterior distribution $p(w \mid \mathcal{D})$ as in Example 2. Let the predictive distribution $p(y \mid x, w)$ be a Gaussian distribution $p_{\mathcal{N}}(y; f_{\boldsymbol{w}}(x), 1)$ with mean $f_{\boldsymbol{w}}(x)$ and variance 1. Given input $x = 1$, the expected prediction (Equation 1) is*

$$\mathbb{E}_{p(y|x=1)}[y] = \int_{\boxempty_\perp} y \cdot p_{\mathcal{N}}(y \mid 0, 1) \cdot \frac{1}{6} \, dy \, dw + \int_{\boxempty_\top} y \cdot p_{\mathcal{N}}(y \mid w, 1) \cdot \frac{1}{6} \, dy \, dw.$$

*It is a summation of two WVC problems with $\boxempty_\perp = (-3 \le w \le 0) \wedge (y \in \mathbb{R})$ and $\boxempty_\top = (0 \le w \le 3) \wedge (y \in \mathbb{R})$, whose joint integral surface is shown in Figure 1a.*

These WVC problems do not admit closed-form solutions since they involve truncated Gaussian distributions. Moreover, Figure 1a shows that computing BMA, even in such a low-dimensional parameter space, requires integration over non-convex and multi-modal functions.

## 2.2 General Reduction of BMA to WVC

Let model $f_{\boldsymbol{w}}$ be a $\mathsf{ReLU}$ neural net. Denote the set of inputs to its $\mathsf{ReLU}$ activations by $\mathcal{R} = \{r_i\}_{i=1}^R$, where each $r_i$ is a linear combination of weights. For a given input $\boldsymbol{x}$, the parameter space is partitioned by whether each $\mathsf{ReLU}$ activation outputs zero or not. This gives the WVC reduction

$$p(y \mid \boldsymbol{x}) = \sum_{\boldsymbol{B} \in \{0,1\}^R} \int_{\boxempty_{\boldsymbol{B}}} p(y \mid \boldsymbol{x}, \boldsymbol{w}) \, p(\boldsymbol{w} \mid \mathcal{D}) \, d\boldsymbol{w},$$

where $\boldsymbol{B}$ is a binary vector. The region $\boxempty_{\boldsymbol{B}}$ is defined as $\wedge_{i=1}^R \ell_i$ where arithmetic constraint $\ell_i$ is $r_i \ge 0$ if $\boldsymbol{B}_i = 1$ and $r_i \le 0$ otherwise. The expected prediction $\mathbb{E}_{p(y|\boldsymbol{x})}[y]$ is analogous but includes an additional factor and variable of integration $y$ in each WVC problem.

This general reduction, however, is undesirable since it amounts to a brute-force enumeration that implies a complexity exponential in the number of $\mathsf{ReLU}$ activations. Moreover, not all WVC

problems resulting from this reduction are amenable to existing solvers. We will therefore appeal to a framework called weighted model integration (WMI) that allows for a compact representation of these WVC problems, and a characterization of their tractability for WMI solvers (Kolb et al., 2019). This inspires us to approximate BMA by first reducing it to WVC problems and further closely approximating those with tractable WMI problems.

## 3 Approximating BMA by WMI

WMI is a modeling and inference framework that supports integration in the presence of logical and arithmetic constraints (Belle et al., 2015a,b). Various WMI solvers have been proposed in recent years (Kolb et al., 2019), ranging from general-purpose ones to others that assume some problem structures to gain scalability. However, even with the reduction from BMA to WVC from the previous section, WMI solvers are not directly applicable. Existing solvers have two main limitations: (i) feasible regions need to be defined by Boolean combinations of linear arithmetic constraints, and (ii) weight functions need to be polynomials. In this section, we show that these issues can be bypassed using a motivating example of how to form a close approximation to BMA using WMI.

In WMI, the feasible region is defined by *satisfiability modulo theories* (SMT) constraints (Barrett et al., 2010): an SMT formula is a (typically quantifier-free) expression containing both propositional and theory literals connected with logical connectives; the theory literals are often restricted to *linear real arithmetic*, where literals are of the form $(\mathbf{c}^T \mathbf{X} \leq b)$ with variable $\mathbf{X}$ and constants $\mathbf{c}^T$ and $b$.

**Example 4.** *The* ReLU *model* $f_{\boldsymbol{w}}(x)$ *of Example 2 can be encoded as an SMT formula (see box).*

*The curly bracket denotes logical conjunction, the symbol $\Rightarrow$ is a logical implication, variable $W$ is the weight, and variable $Z$ denotes the model output.*

$$\Delta_{\mathsf{ReLU}} = \left\{ \begin{array}{l} W \cdot x > 0 \Rightarrow Z = W \cdot x \\ W \cdot x \leq 0 \Rightarrow Z = 0 \end{array} \right.$$

The encoding of ReLU neural networks into SMT formulas is explored in existing work to enable verification of the behavior of neural networks and provide formal guarantees (Katz et al., 2017; Huang et al., 2017; Sivaraman et al., 2020). We propose to use this encoding to define the feasible region of WMI problems. Let $\boldsymbol{x} \models \Delta$ denote the satisfaction of an SMT formula $\Delta$ by an assignment $\boldsymbol{x}$, and $[\![\boldsymbol{x} \models \Delta]\!]$ be its corresponding indicator function. We formally introduce WMI next.

**Definition 5.** *(WMI) Let $\boldsymbol{X}$ be a set of continuous random variables. A weighted model integration problem is a pair $\mathcal{M} = (\Delta, \Phi)$, where $\Delta$ is an SMT formula over $\boldsymbol{X}$ and $\Phi$ is a set of per-literal weights defined as $\Phi = \{\phi_\ell\}_{\ell \in \mathcal{L}}$, where $\mathcal{L}$ is a set of SMT literals and each $\phi_\ell$ is a function defined over variables in literal $\ell$. The task of weighted model integration is to compute*

$$\mathsf{WMI}(\Delta, \Phi) = \int_{\boldsymbol{x} \models \Delta} \prod_{\ell \in \mathcal{L}} \phi_\ell(\boldsymbol{x})^{[\![\boldsymbol{x} \models \ell]\!]} \, d\boldsymbol{x}.$$

That is, the task is to integrate over the weighted assignments of $\boldsymbol{X}$ that satisfy the SMT formula $\Delta$.[1]

An approximation to the BMA of Example 3 can be achieved with WMI using the following four steps:

**Step 1. Encoding model $f_{\boldsymbol{w}}(x)$.** This has been shown as the SMT formula $\Delta_{\mathsf{ReLU}}$ in Example 4.

**Step 2. Encoding posterior distribution $p(w \mid \mathcal{D})$.** The uniform distribution $p(w \mid \mathcal{D}) = \frac{1}{6}$ with $w \in [-3, 3]$ can be encoded as a WMI problem pair $(\Delta_{pos}, \Phi_{pos})$ as follows:

$$\Delta_{pos} = -3 \leq W \leq 3 \qquad \Phi_{pos} = \left\{ \phi_\ell(W) = \frac{1}{6} \ \text{with} \ \ell = \mathtt{true} \right\}$$

**Step 3. Approximate encoding of predictive distribution $p(y \mid w, x)$.** Recall that $p(y \mid w, x) = p_{\mathcal{N}}(y; f_{\boldsymbol{w}}(x), 1)$ is Gaussian, which cannot be handled by existing WMI solvers. To approximate it with polynomial densities, we simply use a triangular distribution encoded as a WMI problem pair:

$$\Delta_{pred} = \left\{ \begin{array}{l} Y \leq Z + \alpha \\ Y \geq Z - \alpha \end{array} \right. \qquad \Phi_{pred} = \left\{ \begin{array}{ll} \phi_{\ell_1}(Y, Z) = \frac{1 - Y + Z}{\alpha} & \text{with} \ \ell_1 = Y \geq Z \\ \phi_{\ell_2}(Y, Z) = \frac{1 + Y - Z}{\alpha} & \text{with} \ \ell_2 = Y < Z \end{array} \right\}$$

---

[1] In the literature, WMI is defined on mixed discrete-continuous domains. Since we only work with WMI problems over continuous variables, we ignore the discrete ones in the definition for succinctness.

In this encoding, $\alpha$ is a constant that defines the shape of the triangular distribution. It is obtained by minimizing the $L2$ distance between a standard normal distribution and the symmetric triangular distribution. We visualize this approximation in the right figure.

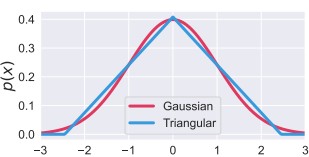

**Step 4. Approximating BMA by calling WMI solvers.** With the above encodings, the predictive posterior $p(y \mid x)$ (Equation 1) can be computed using two calls to a WMI solver. For example, the uncertainty of a prediction $y = 1$ for input $x = 1$ is

$$p(y = 1 \mid x = 1) \ = \ \mathsf{WMI}(\Delta \wedge (Y = 1), \Phi) \ / \ \mathsf{WMI}(\Delta, \Phi) \ = \ 0.164 \ / \ 1,$$

where $\Delta \ = \ \Delta_{\mathsf{ReLU}} \wedge \Delta_{pos} \wedge \Delta_{pred}$ and $\Phi \ = \ \Phi_{pos} \cup \Phi_{pred}$. Similarly, the expected prediction $\mathbb{E}_{p(y|x=1)}[y]$ (Equation 1) can be computed using two calls to a WMI solver:

$$\mathbb{E}_{p(y|x=1)}[y] \ = \ \mathsf{WMI}(\Delta, \Phi^*) \ / \ \mathsf{WMI}(\Delta, \Phi) \ = \ 0.752 \ / \ 1,$$

where $\Phi^* = \Phi \cup \{\phi_\ell(Y) = Y \ \text{with} \ \ell = \mathtt{true}\}$. The above formulations also work for unnormalized distributions since the WMI in the denominator serves to compute the partition function.

We visualize the integral surface of the resulting approximate BMA problem in Figure 1b. It is very close to the integral surface of the original BMA problem in Figure 1a. However, it can be exactly integrated using existing WMI solvers while the original one does not admit such solutions. Next, we show how this process can be generalized to a scalable and accurate approximation of BMA.

## 4 CIBER: Collapsed Inference for Bayesian Deep Learning via WMI

Given a BNN with a large number of weights, naively approximating it by WMI problems can lead to computational issues, since it involves doing integration over polytopes in arbitrarily high dimensions and this is known to be #P-hard (Valiant, 1979; De Loera et al., 2012; Zeng et al., 2020c). Further, weights involved with non-ReLU activation might not be amenable to the WMI encoding. To tackle these issues, we propose to use collapsed samples to combine the strengths from two worlds: the scalability and flexibility from sampling and the accuracy from WMI solvers.

**Definition 6.** *(Collapsed BMA) Let $(W_s, W_c)$ be a partition of parameters $W$. A collapsed sample is a tuple $(w_s, q)$, where $w_s$ is an assignment to the sampled parameters $W_s$ and $q$ is a representation of the conditional posterior $p(W_c \mid w_s, \mathcal{D})$ over the collapsed parameter set $W_c$. Given collapsed samples $\mathcal{S}$, collapsed BMA estimates the predictive posterior and expected prediction as*

$$p(y \mid \boldsymbol{x}) \approx \frac{1}{|\mathcal{S}|} \sum_{(\boldsymbol{w}_s, q) \in \mathcal{S}} \left[ \int p(y \mid \boldsymbol{x}, \boldsymbol{w}) \, q(\boldsymbol{w}_c) \, d\boldsymbol{w}_c \right], \ and$$

$$\mathbb{E}_{p(y|\boldsymbol{x})}[y] \approx \frac{1}{|\mathcal{S}|} \sum_{(\boldsymbol{w}_s, q) \in \mathcal{S}} \left[ \int y \, p(y \mid \boldsymbol{x}, \boldsymbol{w}) \, q(\boldsymbol{w}_c) \, d\boldsymbol{w}_c \, dy \right].$$

$(2)$

The size of the collapsed set $W_c$ determines the trade-off between scalability and accuracy. The more parameters in the collapsed set, the more accurate the approximation to BMA is. The fewer parameters in $W_c$, the more efficient the computations of the integrals are since the integration is performed in a lower-dimensional space. Later in our experiments, we choose a subset of weights at the last or second-to-last hidden layer of the neural networks to be the collapsed set. This choice is known to be effective in capturing uncertainty as shown in Kristiadi et al. (2020); Snoek et al. (2015).

To develop an algorithm to compute collapsed BMA, we are faced with two main design choice questions: **(Q1)** how to sample $w_s$ from the posterior? **(Q2)** what should be the representation of the conditional posterior $q$ such that the integrals in Equation 2 can be computed exactly? Next, we provide our answers to these two questions that together give our proposed algorithm CIBER.

### 4.1 Approximation to Posteriors

For **(Q1)**, we follow Maddox et al. (2019) and sample from the stochastic gradient descent (SGD) trajectory after convergence and use the information contained in SGD trajectories to efficiently approximate the posterior distribution over the parameters of the neural network, leveraging the interpretation of SGD as approximate Bayesian inference (Mandt et al., 2017; Chen et al., 2020).

Given a set of parameter samples $\mathcal{W}$ from the SGD trajectory, the sample set is defined as $\mathcal{W}_s = \{\boldsymbol{w}_s \mid \boldsymbol{w} \in \mathcal{W}\}$. For each assignment $\boldsymbol{w}_s$, an approximation $q(\boldsymbol{W}_c)$ to the conditional posterior $p(\boldsymbol{W}_c \mid \boldsymbol{w}_s, \mathcal{D})$ is necessary since the posteriors induced by SGD trajectories are implicit. Next, we discuss the choice of approximation to the conditional posterior that is amenable to WMI.

## 4.2 Encoding into WMI Problems

As shown in Section 3, if a BNN can be encoded as a WMI problem, the posterior predictive distribution and the expected prediction, which involve marginalization over the parameter space, can be computed exactly using WMI solvers. This inspires us to use the WMI framework as the closed-form representation for the conditional posteriors of parameters. The main challenge is how to approximate the integrand in Equation 2 using an SMT formula and a polynomial weight function in order to obtain a WMI problem amenable to existing solvers.

*For the conditional posterior approximation* $q(\boldsymbol{W}_c)$, we choose it to be a uniform distribution that can be encoded into a WMI problem as $\mathcal{M}_{pos} = (\Delta_{pos}, \Phi_{pos})$ with the SMT formula being $\Delta_{pos} = \wedge_{i \in c} (l_i \leq W_i \leq u_i)$ and weights being $\Phi_{pos} = \{\phi_\ell(\boldsymbol{W}_c) = 1 \mid \ell = \texttt{true}\}$, where $l_i$ and $u_i$ are domain lower and upper bounds for the uniform distribution respectively. While seemingly over-simplistic, this choice of approximation to the conditional posterior is sufficient to robustly deliver surprisingly strong empirical performance as shown in Section 6. The intuition is that uniform distributions are better than a few samples. We further illustrate this point by comparing the predictive distributions of CIBER and HMC in a few-sample setting. Figure 2 shows that even with the same 10 samples drawn from the posterior distribution, since CIBER further approximates the 10 samples with a uniform distribution, it yields a predictive distribution closer to the ground truth than HMC, indicating that using a uniform distribution instead of a few samples forms a better approximation.

*For the choice of predictive distribution* $p(y \mid \boldsymbol{x}, \boldsymbol{w})$, we propose to use piecewise polynomial densities. Common predictive distributions can be approximated by polynomials up to arbitrary precision in theory by the Stone–Weierstrass theorem (De Branges, 1959). For regression, the de facto choice is Gaussian and we propose to use triangular distribution as the approximation, i.e., $p(y \mid \boldsymbol{x}, \boldsymbol{w}) = \frac{1}{r} - \frac{1}{r^2}|y - f_{\boldsymbol{w}}(\boldsymbol{x})|$, with domain $|y - f_{\boldsymbol{w}}(\boldsymbol{x})| \leq r$, and $r := \alpha\sqrt{\sigma^2(\boldsymbol{x})}$ where the constant $\alpha$ parameterizes the triangular distribution as described in Section 3. Here, $\sigma^2(\boldsymbol{x})$ is the variance estimate, which can be a function of input $\boldsymbol{x}$ depending on whether the BNN is homoscedastic or heteroscedastic. Then $p(y \mid \boldsymbol{x}, \boldsymbol{w})$ can be encoded into WMI as:

$$\Delta_{pred} = \left\{ \begin{array}{l} Y - f_{\boldsymbol{w}}(\boldsymbol{x}) \leq r \\ Y - f_{\boldsymbol{w}}(\boldsymbol{x}) \geq -r \end{array} \right. \quad \Phi_{pred} = \left\{ \begin{array}{l} \phi_{\ell_1}(Y, \boldsymbol{W}_c) = \frac{1}{r} - \frac{Y - f_{\boldsymbol{w}}(\boldsymbol{x})}{r^2} \ \text{with} \ \ell_1 = (Y > f_{\boldsymbol{w}}(\boldsymbol{x})) \\ \phi_{\ell_2}(Y, \boldsymbol{W}_c) = \frac{1}{r} - \frac{f_{\boldsymbol{w}}(\boldsymbol{x}) - Y}{r^2} \ \text{with} \ \ell_2 = (f_{\boldsymbol{w}}(\boldsymbol{x}) > Y) \end{array} \right\}$$

Similar piecewise polynomial approximations are adopted for classification tasks when the predictive distributions are induced by softmax functions. Those details are presented in the Appendix.

## 4.3 Exact Integration in Collapsed BMA

By encoding the collapsed BMA into WMI problems, we are ready to answer **(Q2)**, i.e., how to perform exact computation of the integrals shown in Equation 2.

**Proposition 7.** *Let the SMT formula* $\Delta = \Delta_{\textsf{ReLU}} \wedge \Delta_{pos} \wedge \Delta_{pred}$, *and the set of weights* $\Phi = \Phi_{pos} \cup \Phi_{pred}$ *as defined in Section 4.2. Let the set of weights* $\Phi^* = \Phi \cup \{\phi_\ell(Y) = Y \ \text{with} \ \ell = \texttt{true}\}$. *The integrals in collapsed BMA (Equation 2) can be computed by WMI solvers as*

$$\int p(y \mid \boldsymbol{x}, \boldsymbol{w}) \, q(\boldsymbol{w}_c) \, d\boldsymbol{w}_c = \textsf{WMI}(\Delta \wedge (\boldsymbol{Y} = y), \Phi) \, / \, \textsf{WMI}(\Delta, \Phi), \ \text{and}$$

$$\int y \, p(y \mid \boldsymbol{x}, \boldsymbol{w}) \, q(\boldsymbol{w}_c) \, d\boldsymbol{w}_c \, dy = \textsf{WMI}(\Delta, \Phi^*) \, / \, \textsf{WMI}(\Delta, \Phi).$$

With both questions **(Q1)** and **(Q2)** answered, we summarize our proposed algorithm CIBER in Algorithm 1 in the Appendix. To quantitatively analyze how close the approximation delivered by CIBER is to the ground-truth BMA, we consider the following experiments with closed-form BMA.

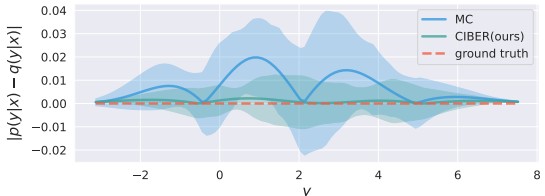
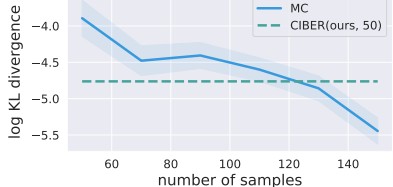

Figure 3: Posterior predictive distributions in Bayesian linear regression. The $y$-axis shows the absolute difference between an estimated predictive distribution $p(y \mid \mathbf{x})$ and the ground-truth predictive distribution $q(y \mid \mathbf{x})$. Shaded regions are the 95% confidence interval. Best viewed in color.

Figure 4: KL divergence in Bayesian linear regression. The $x$-axis shows the number of samples the MC method uses for estimations, ranging from 50 to 150. The blue curve shows the MC method, and the green dashed curve shows CIBER using 50 samples.

**Regression.** We consider a Bayesian linear regression setting where exact sampling from the posterior distribution is available. Both the likelihood and the weight posterior are Gaussian such that the ground-truth posterior predictive distribution is Gaussian as well. With samples drawn from the weight posterior, CIBER approximates the samples with a uniform distribution as posterior $p(\boldsymbol{w}|\mathcal{D})$ and further approximates the likelihood with a triangular distribution such that the integral $p(y|\boldsymbol{x}, \mathcal{D}) = \int p(y|\boldsymbol{x}, \boldsymbol{w})p(\boldsymbol{w}|\mathcal{D}) \, d\boldsymbol{w}$ can be computed exactly by WMI.

We first evaluate the posterior predictive distribution estimated by CIBER and Monte Carlo (MC) method, using the same five samples drawn from the weight posterior. Results averaged over 10 trials are shown in Figure 3 where the estimations by CIBER are much closer to the ground truth posterior predictive distribution than those by the MC method. Further, the averaged KL divergence between the ground truth and CIBER is 0.069 while the one for MC estimations is 0.130, again indicating that CIBER yields a better BMA approximation in the few-sample setting.

We further explore the question of how many samples the MC method needs to match the performance of CIBER. The performances of both approaches are evaluated using KL divergence between the ground-truth posterior distribution and the estimated one, averaged over 10 trials. The result is shown in Figure 4 where the dashed green line shows the performance of CIBER with 50 samples and the blue curve shows the performance of MC with an increasing number of samples. As expected, the MC method yields lower KL divergence as the number of samples increases; however, it takes more than 100 samples to match CIBER, indicating its low sample efficiency and that developing efficient and effective inference algorithms such as CIBER for estimating BMA is a meaningful question.

**Classification.** For analyzing classification performance, Kristiadi et al. (2022) propose to compute the integral $I = \int \sigma(f_*)p_{\mathcal{N}}(f_*) \, df_*$ with $\sigma$ being the sigmoid function and $f_* = f(\boldsymbol{x}^*; \boldsymbol{w})$ that amounts to the posterior predictive distribution. We consider a simple case with $f(\boldsymbol{x}; \boldsymbol{w}) = \boldsymbol{w} \cdot \boldsymbol{x}$ such that the ground-truth integral can be obtained. With a randomly chosen $\boldsymbol{x}$, the ground-truth integral is $I = 0.823$. The integral estimated by CIBER is $I_C = 0.826$ while the MC estimate is $I_{MC} = 0.732$. That is, CIBER gives an estimate with a much lower error than the MC estimation error, indicating that CIBER is able to deliver high-quality approximations in classification tasks.

# 5 Related Work

**Bayesian Deep Learning.** Bayesian inference over deep neural networks (MacKay, 1992) is proposed to fix the issue that deep learning models give poor uncertainty estimations and suffer from overconfidence (Nguyen et al., 2015; Hein et al., 2019; Meronen et al., 2023, 2021). Some methods use samples from SGD trajectories to approximate the implicit true posteriors similar to us: Izmailov et al. (2020) (SI) proposes to perform Bayesian inference in a subspace of the parameter space spanned by a few vectors derived from principal component analysis (PCA+ESS(SI)) or variational inference (PCA+VI(SI)); SWAG (Maddox et al., 2019) proposes to approximate the full parameter space using an approximate Gaussian posterior whose mean and covariance are from a partial SGD trajectory with a modified learning rate scheduler.

Some other approaches using approximate posteriors include MC Dropout (MCD) (Gal & Ghahramani, 2015, 2016) which is one of the Bayesian dropout methods and recently, one of its modifications called Variational Structured Dropout (VSD) (Nguyen et al., 2021) using variational inference is pro-

Table 1: Average test log likelihood for the small UCI regression task.

|  | Boston | Concrete | Yacht | Naval | Energy |
|---|---|---|---|---|---|
| CIBER (second) | **-2.471 ± 0.140** | -2.975 ± 0.102 | -0.678 ± 0.301 | 7.276 ± 0.532 | **-0.716 ± 0.211** |
| CIBER (last) | **-2.471 ± 0.140** | **-2.959 ± 0.109** | -0.687 ± 0.301 | **7.482 ± 0.188** | **-0.716 ± 0.211** |
| SWAG | -2.761 ± 0.132 | -3.013 ± 0.086 | -0.404 ± 0.418 | 6.708 ± 0.105 | -1.679 ± 1.488 |
| PCA+ESS (SI) | -2.719 ± 0.132 | -3.007 ± 0.086 | **-0.225 ± 0.400** | 6.541 ± 0.095 | -1.563 ± 1.243 |
| PCA+VI (SI) | -2.716 ± 0.133 | -2.994 ± 0.095 | -0.396 ± 0.419 | 6.708 ± 0.105 | -1.715 ± 1.588 |
| SGD | -2.752 ± 0.132 | -3.178 ± 0.198 | -0.418 ± 0.426 | 6.567 ± 0.185 | -1.736 ± 1.613 |
| DVI | -2.410 ± 0.020 | -3.060 ± 0.010 | -0.470 ± 0.030 | 6.290 ± 0.040 | -1.010 ± 0.060 |
| DGP | -2.330 ± 0.060 | -3.130 ± 0.030 | -1.390 ± 0.140 | 3.600 ± 0.330 | -1.320 ± 0.030 |
| VI | -2.430 ± 0.030 | -3.040 ± 0.020 | -1.680 ± 0.040 | 5.870 ± 0.290 | -2.380 ± 0.020 |
| MCD | -2.400 ± 0.040 | -2.970 ± 0.020 | -1.380 ± 0.010 | 4.760 ± 0.010 | -1.720 ± 0.010 |
| VSD | -2.350 ± 0.050 | -2.970 ± 0.020 | -1.140 ± 0.020 | 4.830 ± 0.010 | -1.060 ± 0.010 |

posed. Other state-of-the-art approximate BNN inference methods including deterministic variational inference (DVI) (Wu et al., 2019), deep Gaussian processes (DGP) (Bui et al., 2016) with Gaussian process layers and variational inference (VI) (Kingma & Welling, 2013). Closely related to DGP is the deep kernel process (Aitchison et al., 2021) that writes DGPs as deep Wishart processes.

**WMI Solvers.** WMI generalizes weighted model counting (WMC) Sang et al. (2005), a state-of-the-art inference approach in many discrete probabilistic models, from discrete to mixed discrete-continuous domains Belle et al. (2015a,b). Recent research on WMI includes its tractability (Zeng et al., 2020c, 2021; Abboud et al., 2020) and the advancements in WMI solver development. Existing exact WMI solvers for arbitrarily structured problems include DPLL-based search with numerical Belle et al. (2015a); Morettin et al. (2017, 2019) or symbolic integration de Salvo Braz et al. (2016) and compilation-based algorithms Kolb et al. (2018); Zuidberg Dos Martires et al. (2019); Derkinderen et al. (2020) that use extended algebraic decision diagrams (XADDs) (Sanner et al., 2012) as a compilation target which is a powerful tool for inference on mixed domains (Sanner & Abbasnejad, 2012; Zamani et al., 2012). Some exact WMI solvers aiming to improve efficiency for a certain class of models are proposed such as SMI (Zeng & Van den Broeck, 2019) and MP-WMI (Zeng et al., 2020a) which are greatly scalable for WMI problems that satisfy certain structural constraints. Approximate solvers are also proposed including sampling-based ones (Zuidberg Dos Martires et al., 2020) and relaxation-based ones (Zeng et al., 2020c,b). Recent WMI efforts converge in the `pywmi` library (Kolb et al., 2019). The SMT formulas considered in this work can be seen as distributional constraints on continuous domains. There is also plenty of work in neuro-symbolic AI exploring the integration of discrete constraints into neural networks models including the architectures (Ahmed et al., 2022b, 2023) and the loss (Xu et al., 2018; Ahmed et al., 2022c,a).

# 6 Experiments

We conduct experimental evaluations of our proposed approach CIBER [1] on regression and classification benchmarks and compare its performance on uncertainty estimation as well as prediction accuracy with a wide range of baseline methods. More experimental details are presented in the Appendix.

## 6.1 Regression on Small and Large UCI Datasets

We experiment on 5 small UCI datasets: *boston*, *concrete*, *yacht*, *naval* and *energy*. We follow the setup of Izmailov et al. (2020) and use a fully connected network with a single hidden layer and 50 units with ReLU activations. We further experiment on 6 large UCI datasets: *elevators*, *keggdirected*, *keggundirected*, *pol*, *protein* and *skillcraft*. We use a feedforward network with five hidden layers of sizes $[1000, 1000, 500, 50, 2]$ and ReLU activations on all datasets except *skillcraft*. For *skillcraft*, a smaller architecture is adopted with four hidden layers of size $[1000, 500, 50, 2]$. All models have two outputs for the prediction and the heteroscedastic variance respectively.

---

[1]Code and experiments are available at `https://github.com/UCLA-StarAI/CIBER`.

Table 2: Average test log likelihood for the large UCI regression task.

| | ELEVATORS | KEGGD | KEGGU | PROTEIN | SKILLCRAFT | POL |
|---|---|---|---|---|---|---|
| CIBER (SECOND) | -0.378 ± 0.026 | **1.245 ± 0.090** | **1.125 ± 0.269** | -0.720 ± 0.036 | -1.003 ± 0.035 | **2.555 ± 0.115** |
| CIBER (LAST) | -0.371 ± 0.023 | 1.178 ± 0.088 | 0.964 ± 0.231 | -0.720 ± 0.036 | **-1.001 ± 0.032** | 2.506 ± 0.150 |
| SWAG | -0.374 ± 0.021 | 1.080 ± 0.035 | 0.749 ± 0.029 | **-0.700 ± 0.051** | -1.180 ± 0.033 | 1.533 ± 1.084 |
| PCA+ESS (SI) | -0.351 ± 0.030 | 1.074 ± 0.034 | 0.752 ± 0.025 | -0.734 ± 0.063 | -1.181 ± 0.033 | -0.185 ± 2.779 |
| PCA+VI (SI) | **-0.325 ± 0.019** | 1.085 ± 0.031 | 0.757 ± 0.028 | -0.712 ± 0.057 | -1.179 ± 0.033 | 1.764 ± 0.271 |
| SGD | -0.538 ± 0.108 | 1.012 ± 0.154 | 0.602 ± 0.224 | -0.854 ± 0.085 | -1.162 ± 0.032 | 1.073 ± 0.858 |
| ORTHVGP | -0.448 | 1.022 | 0.701 | -0.914 | — | 0.159 |
| NL | -0.698 ± 0.039 | 0.935 ± 0.265 | 0.670 ± 0.038 | -0.884 ± 0.025 | -1.002 ± 0.050 | -2.840 ± 0.226 |

Table 3: Average test performance for image classification tasks on CIFAR-10 and CIFAR-100.

| METRIC | NLL | | ACC | | ECE | |
|---|---|---|---|---|---|---|
| DATASET | CIFAR-10 | CIFAR-100 | CIFAR-10 | CIFAR-100 | CIFAR-10 | CIFAR-100 |
| CIBER | **0.1927 ± 0.0029** | **0.9193 ± 0.0027** | **93.64 ± 0.09** | **74.71 ± 0.18** | 0.0130 ± 0.0011 | **0.0168 ± 0.0025** |
| SWAG | 0.2503 ± 0.0081 | 1.2785 ± 0.0031 | 93.59 ± 0.14 | 73.85 ± 0.25 | 0.0391 ± 0.0020 | 0.1535 ± 0.0015 |
| SGD | 0.3285 ± 0.0139 | 1.7308 ± 0.0137 | 93.17 ± 0.14 | 73.15 ± 0.11 | 0.0483 ± 0.0022 | 0.1870 ± 0.0014 |
| SWA | 0.2621 ± 0.0104 | 1.2780 ± 0.0051 | 93.61 ± 0.11 | 74.30 ± 0.22 | 0.0408 ± 0.0019 | 0.1514 ± 0.0032 |
| SGLD | 0.2001 ± 0.0059 | 0.9699 ± 0.0057 | 93.55 ± 0.15 | 74.02 ± 0.30 | **0.0082 ± 0.0012** | 0.0424 ± 0.0029 |
| KFAC | 0.2252 ± 0.0032 | 1.1915 ± 0.0199 | 92.65 ± 0.20 | 72.38 ± 0.23 | 0.0094 ± 0.0005 | 0.0778 ± 0.0054 |

We run CIBER with two different ways of choosing the collapsed parameter set: *CIBER (last)* chooses all the weights at the last layer to be the collapsed set; *CIBER (second)* chooses three out of all the weights at the second-to-last layer to be the collapsed set. The heuristic we use for choosing the weights is to look into the sampled weights from SGD trajectories to see which ones have the greatest variance. The intuition is that a greater variance indicates that the weight is prone to have greater uncertainty and thus one might want to perform a more accurate inference over it.

**Baselines.** We compare CIBER to the state-of-the-art approximate BNN inference methods. We separate these methods into two categories: those sampling from SGD trajectories as approximate posteriors, which includes SWAG (Maddox et al., 2019), PCA+ESS (SI) and PCA+VI (SI) (Izmailov et al., 2020), vs. those who do not, which includes the SGD baseline, deterministic variational inference (DVI) (Wu et al., 2019), Deep Gaussian Processes (DGP) (Bui et al., 2016), variational inference (VI) (Kingma & Welling, 2013), MC Dropout (MCD) (Gal & Ghahramani, 2015, 2016), and variational structured dropout (VSD) (Nguyen et al., 2021). These methods achieved state-of-the-art performance on the small UCI datasets. We also compare to baselines Bayesian final layers (NL) (Riquelme et al., 2018), deep kernel learning (DKL) (Wilson et al., 2016), orthogonally decoupled variational GPs (OrthVGP) (Salimbeni et al., 2018) and Fastfood approximate kernels (FF) (Yang et al., 2015), which have achieved state-of-the-art performance on the large UCI datasets.

**Results.** We present the test log likelihoods for small UCI datasets in Table 1 and those for large UCI datasets in Table 2. In both tables, the first block summarizes SGD-trajectory sampling-based approaches and the second summarizes the rest. Underlined results are the best among all and bold results are the best among SGD-trajectory sampling-based approaches. From the results, our CIBER has substantially better performance than all others on three out of the five small UCI datasets four out of six large UCI datasets, with comparable performance on the rest, demonstrating that CIBER provides accurate uncertainty estimation. We also present the test rooted-mean-squared error results in Appendix, where CIBER outperforms all other SGD-trajectory sampling-based baselines on four out of five small UCI datasets and four out of six large UCI datasets; it outperforms all baselines on two small UCI datasets and one large UCI datasets and has comparable performance on the rest. This further illustrates that exact marginalization over conditional approximate posteriors enabled by WMI solvers achieves accurate estimation of the true BMA and boosts predictive performance.

## 6.2 Image Classification

**CIFAR datasets.** We experiment with two image datasets: CIFAR-10 and CIFAR-100 (Krizhevsky et al., 2009) and evaluate the test performance using three metrics: 1) negative log likelihood (NLL)

Table 4: Average test performance for image transfer learning tasks.

| METRIC | NLL | | ACC | | ECE | |
|---|---|---|---|---|---|---|
| MODEL | VGG-16 | PRERESNET-164 | VGG-16 | PRERESNET-164 | VGG-16 | PRERESNET-164 |
| CIBER | **0.9869 ± 0.0102** | **0.9684 ± 0.0075** | **72.56 ± 0.23** | 75.70 ± 0.17 | **0.0925 ± 0.0028** | **0.0704 ± 0.0031** |
| SWAG | 1.3425 ± 0.0015 | 1.3842 ± 0.0122 | 72.30 ± 0.11 | **76.30 ± 0.06** | 0.1988 ± 0.0028 | 0.1668 ± 0.0006 |
| SGD | 1.6528 ± 0.0390 | 1.4790 ± 0.0000 | 72.42 ± 0.07 | 75.56 ± 0.00 | 0.2149 ± 0.0027 | 0.1758 ± 0.0000 |
| SWA | 1.3993 ± 0.0502 | 1.3552 ± 0.0000 | 71.92 ± 0.01 | 76.02 ± 0.00 | 0.2082 ± 0.0056 | 0.1739 ± 0.0000 |

that reflects the quality of both uncertainty estimation and prediction accuracy, 2) classification accuracy (ACC), and 3) expected calibration errors (ECE) (Naeini et al., 2015) that show the difference between predictive confidence and accuracy and should be close to zero for a well-calibrated approach.

We run CIBER by choosing the collapsed parameter set to be 10 weights and 100 weights at the last layer of the neural network models for CIFAR-10 and CIFAR-100 respectively. The weights are chosen using the same heuristic as the one for regression tasks, i.e., to choose the weights whose samples from the SGD trajectories have large variances. We compare CIBER with strong baselines including SWAG (Maddox et al., 2019) reproduced by their open-source implementation, standard SGD, SWA (Izmailov et al., 2018), SGLD (Welling & Teh, 2011) and KFAC (Ritter et al., 2018).

**Transfer from CIFAR-10 to STL-10.** We further consider a transfer learning task using the model trained on CIFAR-10 to be evaluated on dataset STL-10 (Coates et al., 2011). STL-10 shares nine out of ten classes with the CIFAR-10 dataset but has a different image distribution. It is a common benchmark in transfer learning to adapt models trained on CIFAR-10 to STL-10.

**Results.** We present the test classification performance on dataset CIFAR-10 and CIFAR-100 in Table 3 and that of transfer learning in Table 4. The neural network models used in the classification task are VGG-16 networks and the models used in the transfer learning task are VGG-16 and PreResNet-164. More results using different network architectures are presented in the Appendix. With the same number of samples as SWAG, CIBER outperforms SWAG and other baselines in most evaluations and delivers comparable performance otherwise, demonstrating the effectiveness of using collapsed samples in improving uncertainty estimation as well as classification performance.

# 7 Conclusions And Future Work

We reveal the connection between BMA, a way to perform Bayesian deep learning and WVC, which inspires us to approximate BMA using the framework of WMI. To further make this approximation scalable and flexible, we combine it with collapsed samples which gives our algorithm CIBER. CIBER compares favorably to Bayesian deep learning baselines on regression and classification tasks. A future direction would be to explore what other layers can be expressed as SMT formulas and thus amenable to SMT encoding. Also, the current WMI solvers are limited to polynomial weights, and thus the reduction to WMI problems is applicable to piecewise polynomial weights. This limitation might be alleviated in the future by the development of new WMI solvers that allow various weight function families.

## Acknowledgments

We would like to thank Yuhang Fan and Kareem Ahmed for helpful discussions. We would also like to thank Wesley Maddox for answering queries on the implementation of baseline SWAG. This work was funded in part by the DARPA PTG Program under award HR00112220005, the DARPA ANSR program under award FA8750-23-2-0004, NSF grants #IIS-1943641, #IIS-1956441, #CCF-1837129, and a gift from RelationalAI. GVdB discloses a financial interest in RelationalAI. ZZ is supported by an Amazon Doctoral Student Fellowship.

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

# A  Proofs

**Proposition 7** *Let the SMT formula $\Delta = \Delta_{\mathsf{ReLU}} \wedge \Delta_{pos} \wedge \Delta_{pred}$, and the set of weights $\Phi = \Phi_{pos} \cup \Phi_{pred}$ as defined in Section 4.2. Let the set of weights $\Phi^* = \Phi \cup \{\phi_\ell(Y) = Y \text{ with } \ell = \mathtt{true}\}$. The integrals in collapsed BMA (Equation 2) can be computed by WMI solvers as*

$$\int p(y \mid \boldsymbol{x}, \boldsymbol{w}) \, q(\boldsymbol{w}_c) \, d\boldsymbol{w}_c = \mathsf{WMI}(\Delta \wedge (\boldsymbol{Y} = y), \Phi) \, / \, \mathsf{WMI}(\Delta, \Phi), \text{ and}$$

$$\int y \, p(y \mid \boldsymbol{x}, \boldsymbol{w}) \, q(\boldsymbol{w}_c) \, d\boldsymbol{w}_c \, dy = \mathsf{WMI}(\Delta, \Phi^*) \, / \, \mathsf{WMI}(\Delta, \Phi).$$

*Proof.* By construction, it holds that

$$p(y \mid \boldsymbol{x}, \boldsymbol{w}) \propto \mathsf{WMI}(\Delta_{pred} \wedge (\boldsymbol{Y} = y), \Phi_{pred})$$
$$\propto \prod_{\ell \in \mathcal{L}_{pred}} \phi_\ell(y, \boldsymbol{w}_c)^{[\![y, \boldsymbol{w}_c \models \ell]\!]}, \text{ with } (y, \boldsymbol{w}_c) \models \Delta_{pred} \wedge \Delta_{\mathsf{ReLU}}$$
$$q(\boldsymbol{w}_c) \propto \mathsf{WMI}(\Delta_{pos} \wedge (\boldsymbol{W}_c = \boldsymbol{w}_c), \Phi_{pos})$$
$$\propto \prod_{\ell \in \mathcal{L}_{pos}} \phi_\ell(\boldsymbol{w}_c)^{[\![\boldsymbol{w}_c \models \ell]\!]}, \text{ with } \boldsymbol{w}_c \models \Delta_{pos}$$

Thus, we have that the likelihood weighted by the approximate posterior would be

$$p(y \mid \boldsymbol{x}, \boldsymbol{w}) \, q(\boldsymbol{w}_c) \propto \prod_{\ell \in \mathcal{L}_{pred} \wedge \mathcal{L}_{pos}} \phi_\ell(y, \boldsymbol{w}_c)^{[\![y, \boldsymbol{w}_c \models \ell]\!]} \text{ with } (y, \boldsymbol{w}_c) \models \Delta_{\mathsf{ReLU}} \wedge \Delta_{pred} \wedge \Delta_{pos},$$

or equivalently,

$$p(y \mid \boldsymbol{x}, \boldsymbol{w}) \, q(\boldsymbol{w}_c) = \frac{\prod_{\ell \in \mathcal{L}_{pred} \wedge \mathcal{L}_{pos}} \phi_\ell(y, \boldsymbol{w}_c)^{[\![y, \boldsymbol{w}_c \models \ell]\!]}}{\mathsf{WMI}(\Delta, \Phi)}, \text{ with } (y, \boldsymbol{w}_c) \models \Delta.$$

By integrating over the collapsed set $\boldsymbol{W}_c$, it further holds that

$$\int p(y \mid \boldsymbol{x}, \boldsymbol{w}) \, q(\boldsymbol{w}_c) \, d\boldsymbol{w}_c$$
$$= \frac{\int \prod_{\ell \in \mathcal{L}_{pred} \wedge \mathcal{L}_{pos}} \phi_\ell(y, \boldsymbol{w}_c)^{[\![y, \boldsymbol{w}_c \models \ell]\!]} \, d\boldsymbol{w}_c}{\mathsf{WMI}(\Delta, \Phi)}, \text{ with } (y, \boldsymbol{w}_c) \models \Delta$$
$$= \frac{\mathsf{WMI}(\Delta \wedge (\boldsymbol{Y} = y), \Phi)}{\mathsf{WMI}(\Delta, \Phi)}$$

which proves the first equation.

Similarly, we have that

$$y \, p(y \mid \boldsymbol{x}, \boldsymbol{w}) \, q(\boldsymbol{w}_c)$$
$$\propto \prod_{\ell \in \mathcal{L}_{pred}} y \, \phi_\ell(y, \boldsymbol{w}_c)^{[\![y, \boldsymbol{w}_c \models \ell]\!]} \prod_{\ell \in \mathcal{L}_{pos}} \phi_\ell(\boldsymbol{w}_c)^{[\![\boldsymbol{w}_c \models \ell]\!]}, \text{ with } (y, \boldsymbol{w}_c) \models \Delta$$

By integrating over the collapsed set $\boldsymbol{W}_c$ and prediction $y$, it holds that

$$\int y \, p(y \mid \boldsymbol{x}, \boldsymbol{w}) \, q(\boldsymbol{w}_c) \, d\boldsymbol{w}_c \, dy$$
$$= \frac{\int y \prod_{\ell \in \mathcal{L}_{pred}} \phi_\ell(y, \boldsymbol{w}_c)^{[\![y, \boldsymbol{w}_c \models \ell]\!]} \prod_{\ell \in \mathcal{L}_{pos}} \phi_\ell(\boldsymbol{w}_c)^{[\![\boldsymbol{w}_c \models \ell]\!]} \, d\boldsymbol{w}_c \, dy}{\mathsf{WMI}(\Delta, \Phi)}, \text{ with } (y, \boldsymbol{w}_c) \models \Delta$$
$$= \frac{\mathsf{WMI}(\Delta, \Phi^*)}{\mathsf{WMI}(\Delta, \Phi)}$$

which finishes our proof. $\qquad \square$

**Algorithm 1** CIBER

**Input**: input $\boldsymbol{x}$, sampled weights $\mathcal{W}$, neural network model $f_{\boldsymbol{w}}$, prediction ground truth $y^*$
**Ouput**: predictions and likelihoods

1: Choose a partition $(\boldsymbol{W}_s, \boldsymbol{W}_c)$ for network parameters
2: Derive approximate posterior $q(\boldsymbol{w}_c)$ from sampled weights $\{\boldsymbol{w}_c \mid \boldsymbol{w} \in \mathcal{W}\}$    *// cf. Section 4.2*
3: Encode posterior $q(\boldsymbol{w}_c)$ into WMI problem $\mathcal{M}_{pos} = (\Delta_{pos}, \Phi_{pos})$    *// cf. Section 4.2*
4: $\mathcal{Y} \leftarrow \emptyset, \mathcal{P} \leftarrow \emptyset$    *// Initialization*
5: **for** sample $\boldsymbol{w}_s$ in $\{\boldsymbol{w}_s \mid \boldsymbol{w} \in \mathcal{W}\}$ **do**
6:      Encode neural network model $f_{\mathsf{ReLU}}$ parameterized by $(\boldsymbol{w}_s, \boldsymbol{W}_c)$ into an SMT formula $\Delta_{f_{\boldsymbol{w}}}$
7:      Encode predictive $p(Y \mid \boldsymbol{x}, \boldsymbol{w}_s, \boldsymbol{W}_c)$ into a WMI problem $\mathcal{M}_{pred} = (\Delta_{pred}, \Phi_{pred})$
8:      SMT formula $\Delta \leftarrow \Delta_{\mathsf{ReLU}} \wedge \Delta_{pos} \wedge \Delta_{pred}$
9:      Weights $\Phi \leftarrow \Phi_{pos} \cup \Phi_{pred}$
10:     Weights $\Phi^* \leftarrow \Phi \cup \{\phi_\ell(Y) = Y \ \text{with} \ \ell = \mathtt{true}\}$
11:     Add prediction $y = \mathsf{WMI}(\Delta, \Phi^*)/\mathsf{WMI}(\Delta, \Phi)$ to prediction set $\mathcal{Y}$    *// cf. Section 4.3*
12:     Add likelihood $p = \mathsf{WMI}(\Delta \wedge (Y = y^*), \Phi)/\mathsf{WMI}(\Delta, \Phi)$ to set $\mathcal{P}$    *// cf. Section 4.3*
13: **return** $y = \mathrm{MEAN}(\mathcal{Y}), p(y^* \mid \boldsymbol{x}) = \mathrm{MEAN}(\mathcal{P})$

Table 5: Average test RMSE for the small UCI regression task.

| | BOSTON | CONCRETE | YACHT | NAVAL | ENERGY |
|---|---|---|---|---|---|
| CIBER (SECOND) | $3.488 \pm 1.123$ | $4.880 \pm 0.506$ | $0.828 \pm 0.241$ | $\mathbf{0.000 \pm 0.000}$ | $\mathbf{0.447 \pm 0.081}$ |
| CIBER (LAST) | $3.478 \pm 1.128$ | $\mathbf{4.854 \pm 0.503}$ | $\mathbf{0.752 \pm 0.294}$ | $\mathbf{0.000 \pm 0.000}$ | $\mathbf{0.447 \pm 0.081}$ |
| SWAG | $3.517 \pm 0.981$ | $5.233 \pm 0.417$ | $0.973 \pm 0.375$ | $0.001 \pm 0.000$ | $1.594 \pm 0.273$ |
| PCA+ESS (SI) | $3.453 \pm 0.953$ | $5.194 \pm 0.448$ | $0.972 \pm 0.375$ | $0.001 \pm 0.000$ | $1.598 \pm 0.274$ |
| PCA+VI (SI) | $\mathbf{3.457 \pm 0.951}$ | $5.142 \pm 0.418$ | $0.973 \pm 0.375$ | $0.001 \pm 0.000$ | $1.587 \pm 0.272$ |
| SGD | $3.504 \pm 0.975$ | $5.194 \pm 0.446$ | $0.973 \pm 0.374$ | $0.001 \pm 0.000$ | $1.602 \pm 0.275$ |
| MCD | $2.830 \pm 0.170$ | $4.930 \pm 0.140$ | $0.720 \pm 0.050$ | $\underline{0.000 \pm 0.000}$ | $1.080 \pm 0.030$ |
| VSD | $\underline{2.640 \pm 0.170}$ | $\underline{4.720 \pm 0.110}$ | $\underline{0.690 \pm 0.060}$ | $\underline{0.000 \pm 0.000}$ | $0.470 \pm 0.010$ |

## B   Pseudo Code for CIBER

We summarize our proposed algorithm **CIBER**, **C**ollapsed **I**nference **B**ayesian D**E**ep Lea**R**ning, for regression tasks, in Algorithm 1. For the classification task, the algorithm is basically the same except the encoding of the predictive of the distribution. Specifically, for a given class $y$, the predictive distribution $p(y \mid \boldsymbol{x}, \boldsymbol{w})$ can be encoded into a WMI problem as shown below:

$$\Delta_{pred} = \ f_{\boldsymbol{w}}(\boldsymbol{x}) \geq -d \quad \Phi_{pred} = \left\{ \begin{array}{ll} \phi_{\ell_1}(\boldsymbol{W}_c) & with \ \ell_1 = (f_{\boldsymbol{w}}(\boldsymbol{x}) \leq d) \\ \phi_{\ell_2}(\boldsymbol{W}_c) = 1 & with \ \ell_2 = (f_{\boldsymbol{w}}(\boldsymbol{x}) > d) \end{array} \right\}$$

where $\phi_{\ell_1}$ is a cubic polynomial that approximates the sigmoid function such that the posterior predictive distribution $p(y \mid \boldsymbol{x})$ can be solved by WMI solvers by $p(y \mid \boldsymbol{x}) = \mathsf{WMI}(\Delta, \Phi)$. Further, the prediction of BMA for classification tasks is made by $y^* = \arg\max_y p(y \mid \boldsymbol{x})$.

## C   Additional Experiments

### C.1   Toy Regression in Figure 2

We evaluate the predictive distributions obtained by our CIBER and HMC respectively, in a toy dataset generated by sampling 10 input $x$ uniformly distributed in the interval $[-1, -0.5]$ and interval $[0.5, 1]$. For each input $x$, the corresponding target $y$ is computed from a cubic polynomial with Gaussian noises. We apply to these data a Bayesian neural network which is a ReLU neural network with two hidden layers, where both parameter priors and likelihood are Gaussian distributions. We compare HMC and our CIBER in a few-sample setting which is common in most Bayesian deep learning applications, with 10 samples from the posterior distribution. An estimation generated by HMC with a sufficiently large number of samples of size $2,000$ is further presented as a ground truth.

Table 6: Average test RMSE for the large UCI regression task.

| | ELEVATORS | KEGGD | KEGGU | PROTEIN | SKILLCRAFT | POL |
|---|---|---|---|---|---|---|
| CIBER (SECOND) | **0.088 ± 0.002** | 0.142 ± 0.074 | **0.115 ± 0.007** | 0.438 ± 0.009 | **0.251 ± 0.010** | 2.212 ± 0.230 |
| CIBER (LAST) | **0.088 ± 0.002** | 0.142 ± 0.072 | 0.118 ± 0.012 | 0.438 ± 0.009 | **0.251 ± 0.010** | **2.199 ± 0.182** |
| SWAG | **0.088 ± 0.001** | 0.129 ± 0.029 | 0.160 ± 0.043 | 0.415 ± 0.018 | 0.293 ± 0.015 | 3.110 ± 0.070 |
| PCA+ESS (SI) | 0.089 ± 0.002 | 0.129 ± 0.028 | 0.160 ± 0.043 | 0.425 ± 0.017 | 0.293 ± 0.015 | 3.755 ± 6.107 |
| PCA+VI (SI) | **0.088 ± 0.001** | **0.128 ± 0.028** | 0.160 ± 0.043 | 0.418 ± 0.021 | 0.293 ± 0.015 | 2.499 ± 0.684 |
| SGD | 0.103 ± 0.035 | 0.132 ± 0.017 | 0.186 ± 0.034 | 0.436 ± 0.011 | 0.288 ± 0.014 | 3.900 ± 6.003 |
| NL | 0.101 ± 0.002 | 0.134 ± 0.036 | 0.120 ± 0.003 | 0.447 ± 0.012 | 0.253 ± 0.011 | 4.380 ± 0.853 |
| DKL | 0.084 ± 0.020 | 0.100 ± 0.010 | 0.110 ± 0.000 | 0.460 ± 0.010 | 0.250 ± 0.000 | 6.617 |
| ORTHVGP | 0.095 | 0.120 | 0.117 | 0.461 | — | 4.300 ± 0.200 |
| FF | 0.089 ± 0.002 | 0.120 ± 0.000 | 0.120 ± 0.000 | 0.470 ± 0.010 | 0.250 ± 0.020 | — |

Table 7: Average test log likelihoods for image classification tasks on CIFAR-10 and CIFAR-100.

| | CIFAR-10 | | | CIFAR-100 | | |
|---|---|---|---|---|---|---|
| MODEL | VGG-16 | PRERESNET-164 | WIDERESNET | VGG-16 | PRERESNET-164 | WIDERESNET |
| CIBER | **0.1927 ± 0.0029** | **0.1352 ± 0.0014** | 0.1913 ± 0.0029 | **0.9193 ± 0.0027** | 0.8144 ± 0.0065 | 0.7930 ± 0.0065 |
| SWAG | 0.2503 ± 0.0081 | 0.1459 ± 0.0013 | 0.1076 ± 0.0009 | 1.2785 ± 0.0031 | 1.0703 ± 0.4861 | 0.6719 ± 0.0035 |
| SGD | 0.3285 ± 0.0139 | 0.1814 ± 0.0025 | 0.1294 ± 0.0022 | 1.7308 ± 0.0137 | 0.9465 ± 0.0191 | 0.7958 ± 0.0089 |
| SWA | 0.2621 ± 0.0104 | 0.1450 ± 0.0042 | **0.1075 ± 0.0004** | 1.2780 ± 0.0051 | 0.7370 ± 0.0265 | **0.6684 ± 0.0034** |
| SGLD | 0.2001 ± 0.0059 | 0.1418 ± 0.0005 | 0.1289 ± 0.0009 | 0.9699 ± 0.0057 | **0.6981 ± 0.0052** | 0.6780 ± 0.0022 |
| KFAC | 0.2252 ± 0.0032 | 0.1471 ± 0.0012 | 0.1210 ± 0.0020 | 1.1915 ± 0.0199 | 0.7881 ± 0.0025 | 0.7692 ± 0.0092 |

The results are shown in Figure 2. Even with the same 10 samples drawn from the posterior distribution, since CIBER further approximates the 10 samples with a uniform distribution as $q(\boldsymbol{w})$, it yields a predictive distribution $p(y \mid \boldsymbol{x})$ closer to the ground truth than HMC. The intuition behind is that using a uniform distribution instead of a few samples forms a better approximation to the true posterior since the uniform distribution in a collapsed sample represents uncountably many models.

## C.2 Regression on Small and Large Datasets

**Sampling from SGD Trajectories.** During training, we use Gaussian log likelihood as the objective for obtaining smooth gradients and use early stopping to prevent over-fitting. At convergence, we start the sampling process by keeping running SGD and collecting the weights. At deployment time, we approximate the Gaussian predictive distribution with the triangular distributions.

**Hyperparameters.** The hyperparameters including learning rates and weight decay are tuned by performing a grid search to maximize the Gaussian log likelihood using a validation split.

**Additional Results.** Following the set-up of Izmailov et al. (2020), the test performance on small UCI datasets are averaged over 20 trials and the test performance of large UCI datasets are averaged over 10 trials. The test log likelihood results for small UCI datasets are unnormalized while those for large UCI datasets are normalized. Besides the log likelihood results as shown in Section 6.1, root-mean-squared-error (RMSE) results for small UCI datasets are further presented in Table 5 and RMSE results for large UCI datasets are presented in Table 6.

## C.3 Image Classification

**Sampling from SGD Trajectories.** All the network models are trained for 300 epochs using SGD. We start the weight collection after epoch 160 with step size 5. We follow exactly the same hyperparameters as Maddox et al. (2019) including learning rates and weight decay parameters.

**Additional Results.** Following the set-up of Maddox et al. (2019), we run experiments with VGG-16, PreResNet-164 and WideResNet network models on both the image classification task and the transfer learning task. For the image classification task on CIFAR datasets, we present the log likelihood results in Table 7, the accuracy results in Table 8, and ECE results in Table 9. For the transfer learning task from dataset CIFAR-10 to dataset STL-10, we present the results for model VGG-16 and PreResNet-164 in Table 4 and the results for model WideResNet in Table 10.

Table 8: Average test accuracy for image classification tasks on CIFAR-10 and CIFAR-100.

| | CIFAR-10 | | | CIFAR-100 | | |
|---|---|---|---|---|---|---|
| MODEL | VGG-16 | PRERESNET-164 | WIDERESNET | VGG-16 | PRERESNET-164 | WIDERESNET |
| CIBER | **93.64 ± 0.09** | 95.95 ± 0.06 | 95.63 ± 0.16 | **74.71 ± 0.18** | 79.23 ± 0.25 | 81.25 ± 0.35 |
| SWAG | 93.59 ± 0.14 | **96.09 ± 0.08** | 96.38 ± 0.08 | 73.85 ± 0.25 | 73.02 ± 10.30 | 82.27 ± 0.07 |
| SGD | 93.17 ± 0.14 | 95.49 ± 0.06 | 96.41 ± 0.10 | 73.15 ± 0.11 | 78.50 ± 0.32 | 80.76 ± 0.29 |
| SWA | 93.61 ± 0.11 | **96.09 ± 0.08** | **96.46 ± 0.04** | 74.30 ± 0.22 | **80.19 ± 0.52** | **82.40 ± 0.16** |
| SGLD | 93.55 ± 0.15 | 95.55 ± 0.04 | 95.89 ± 0.02 | 74.02 ± 0.30 | 80.09 ± 0.05 | 80.94 ± 0.17 |
| KFAC | 92.65 ± 0.20 | 95.49 ± 0.06 | 96.17 ± 0.00 | 72.38 ± 0.23 | 78.51 ± 0.05 | 80.94 ± 0.41 |

Table 9: Average test ECE for image classification tasks on CIFAR-10 and CIFAR-100.

| | CIFAR-10 | | | CIFAR-100 | | |
|---|---|---|---|---|---|---|
| MODEL | VGG-16 | PRERESNET-164 | WIDERESNET | VGG-16 | PRERESNET-164 | WIDERESNET |
| CIBER | 0.0130 ± 0.0011 | 0.0250 ± 0.0005 | 0.0760 ± 0.0011 | **0.0168 ± 0.0025** | 0.1423 ± 0.0029 | 0.1650 ± 0.0046 |
| SWAG | 0.0391 ± 0.0020 | 0.0214 ± 0.0005 | 0.0096 ± 0.0006 | 0.1535 ± 0.0015 | 0.1031 ± 0.0471 | 0.0678 ± 0.0006 |
| SGD | 0.0483 ± 0.0022 | 0.0255 ± 0.0009 | 0.0166 ± 0.0007 | 0.1870 ± 0.0014 | 0.1012 ± 0.0009 | 0.0479 ± 0.0010 |
| SWA | 0.0408 ± 0.0019 | 0.0203 ± 0.0010 | 0.0087 ± 0.0002 | 0.1514 ± 0.0032 | 0.0700 ± 0.0056 | 0.0684 ± 0.0022 |
| SGLD | **0.0082 ± 0.0012** | 0.0251 ± 0.0012 | 0.0192 ± 0.0007 | 0.0424 ± 0.0029 | 0.0363 ± 0.0008 | **0.0296 ± 0.0008** |
| KFAC | 0.0094 ± 0.0005 | **0.0092 ± 0.0018** | **0.0060 ± 0.0003** | 0.0778 ± 0.0054 | **0.0158 ± 0.0014** | 0.0379 ± 0.0047 |

Table 10: Average test performance for image transfer learning tasks using WideResNet.

| METRIC | NLL | ACC | ECE |
|---|---|---|---|
| CIBER | **0.8259 ± 0.0148** | 75.02 ± 0.31 | **0.0336 ± 0.0009** |
| SWAG | 1.0142 ± 0.0032 | 76.96 ± 0.08 | 0.1303 ± 0.0008 |
| SGD | 1.1308 ± 0.0000 | 76.75 ± 0.00 | 0.1561 ± 0.0000 |
| SWA | 1.0047 ± 0.0000 | **77.50 ± 0.00** | 0.1413 ± 0.0000 |

