# OpenReview forum: "Collapsed Inference for Bayesian Deep Learning"
_NeurIPS.cc/2023/Conference — NeurIPS 2023 poster_

### Official Review · Reviewer_cv4P · 2023-06-29

**Soundness:** 2 fair
**Presentation:** 3 good
**Contribution:** 3 good
**Rating:** 7
**Confidence:** 4

**Summary:**

The paper presents a new method for calculating Bayesian integrals such as the Bayesian model average (BMA) based on volume computation schemes. Specifically, the authors draw inspiration from a weighted volume computation (WVC) problem. Since the WVC is intractable for common neural networks they approximate it with weighted model integration (WMI) that can compute volumes of literals connected with logical connectivities. Some approximations are used to accommodate WMI solvers, such as approximating a Gaussian likelihood with a triangular distribution, and using the WMI solver on part of the network (usually a few dozens of parameters). Results show superior performance compared to sgd-based Bayesian methods and comparable results to Bayesian last-layer methods.

**Strengths:**

Overall I liked the idea of the paper and the novel view it provides for computing Bayesian integrals. Strengths:
* Novel and interesting characterization of the BMA as a volume computation problem
* Novel way of approximating the BMA and other integrals using WMI
* The authors suggested practical and efficient ways to accommodate WMI to the Bayesian learning framework
* The use of a motivating example throughout is a good idea
* For the most part the paper is written clearly

**Weaknesses:**

There are several issues I would like the authors to address:

**Method.**
* The assumption of a uniform posterior is a bit odd and I didn't understand the intuition as presented in L199 in the main text and L510 in the appendix. It is very reasonable to assume that the posterior is not uniform, even when evaluated only on part of the network. A proper justification or intuition for this choice is missing. Also, the good empirical results can be a byproduct of being Bayesian on only part of the network as evident in [1].

**Experiments.**
* In my opinion, the authors exaggerate in how they present their method performance (e..g, "significant improvements", "new state of the art"). First, the method is not compared against all relevant Bayesian methods so it cannot be considered as SoTA. Second, the improvements are generally mild, and when factoring Bayesian last-layer methods (which are for some reason presented in the Appendix), the method is only comparable to them. Furthermore, with more modern NNs CIBER performance is usually inferior to several baseline methods.
* It is not clear why the authors didn't present the performance of Bayesian last layer methods on the CIFAR datasets. In my opinion, it's as important as presenting SGD-based Bayesian methods.
 * A quantification of the method complexity (e.g., wall-clock time) is missing. To me, it is not clear how expensive WMI solvers are and how it grows with the number of parameters. I would have expected to see a comparison to baseline methods w.r.t that aspect as well.
* Experimental details are severely lacking (e.g., was there a validation split? What were the hyper-parameters? Did you do a grid search over hyper-parameters? If so, on what values? ).
* Information about how to choose some of the constants such as $\alpha, l_i, u_i$ is missing. Have you searched for possible values? In general, how did you set their values?
* Code wasn't added to the submission. When factoring this bullet and the last 2 bullets, in my opinion, the results in this paper are not reproducible and this harms this paper's score.

**Clarity of presentation.**
* I think a short background on the main concepts of SMT will make the paper self-contained and easier to read.
* In Fig. 2 it would help to plot the training points as well.

[1] Sharma, M., Farquhar, S., Nalisnick, E., & Rainforth, T. (2023, April). Do Bayesian Neural Networks Need To Be Fully Stochastic?. In International Conference on Artificial Intelligence and Statistics (pp. 7694-7722). PMLR.

**Questions:**

* Perhaps I do not understand something, since some SMT formulas are built from logical conjunction with conflicting conditions, such as the Relu one, how can they be satisfied? There is no $W$ that satisfies both conditions for the same $x$, no?
* Why did you approximate the Gaussian likelihood with a triangular distribution instead of a truncated Taylor series for the exponential function?
* Have you tested your method's sensitivity to temperature scaling?

**Limitations:**

The authors did not address the limitations of their method. Please see the weaknesses/questions sections.

---

> ### Author Rebuttal · Authors · 2023-08-10
>
> We would like to thank the reviewer for appreciating our work for a novel view of computing Bayesian integrals, a practical and efficient framework, and a clear presentation. In what follows, we will address your concerns, with the references put in the general response due to the character limit.
>
> [uniform approximation to posterior]
> Please see our answer in the global response.
>
> [fully/partially Bayesian]
> The idea of partially stochastic neural networks in [4] is to partition the weights into two sets: a deterministic set where the weights are assigned to be MAP estimation and a stochastic set where the weights are stochastic and approximated using existing Bayesian deep learning methods such as Laplace approximation, variational inference or SWAG. This is different from CIBER where the neural network is fully stochastic instead of being partly stochastic. CIBER also partitions the weights into two sets: a sampling set where weights are approximated by SWAG and a collapsed set where the weights are approximated using our proposed closed-form inference. A future research direction can be to combine both by applying the closed-form inference to the stochastic set in the partially stochastic neural networks to study if the observations on stochasticity in [4] still hold.
>
> [baselines]
> In the empirical evaluations, we compare with baselines ranging from the mostly related ones using samples from SGD trajectories for posterior approximations to variational-inference-based ones, deep ensemble, MC dropout, etc, with strictly more baselines than [8]. The Bayesian final layer method [7] is included in both the main paper in Table 2 for log-likelihood and in the appendix in Table 6 for RMSE as direct comparisons to the numbers reported in [8], where the Bayesian final layer method is only reported in regression tasks and not the classification tasks.
>
>
> [runtime complexity]
> We provide the runtime of CIBER on dataset CIFAR-100 dataset in the classification experiment with the three neural network models being VGG-16, PreResNet-164, and WideResNet28x10 respectively as below, to showcase the empirical computational complexity of CIBER. We compare our runtime with SWAG which is known to be a simple and scalable Bayesian deep learning approach summarized in the table below. The runtime of CIBER consists of two main processes: 1) training time, which is exactly the same as the runtime of SWAG, and 2) WMI solving time. The table shows that CIBER is almost as efficient as SWAG and that WMI solving brings little computational overhead due to the efficiency of the WMI solvers.
>
> | MODEL           |   VGG-16   | PreResNet-164 | WideResNet28x10 |
> |-----------------|:----------:|:-------------:|:---------------:|
> | Training / SWAG | 2h3m59s    | 6h3m18s       | 29h39m28s       |
> | WMI Solving     | 11m43.134s | 11m20.638s    | 19m1.076s       |
>
>
> [experimental details/code]
> We put the experimental details of both the regression and the classification experiments in Section C.2 and Section C.3 respectively. The way that the hyper-parameters including learning rates and weight decays are chosen exactly follows [3] as mentioned in Appendix. The hyper-parameter $\alpha$ as described from Line 142 to Line 145 is 2.45. The parameters $\ell, u$ are the bounds of the uniform posterior defined by the minimum and the maximum of the weight samples drawn from the SGD trajectories respectively. We’ll include these details in the next version to make it more self-contained. We’ve also provided our code using an anonymous link and sent the link to AC for your reference.
>
> [presentation]
> We’ll include the suggested changes in the presentation in the next version.
>
> [Q1] An assignment $\mathbf{x}$ satisfies an SMT formula $\Delta$ defined over variables $\mathbf{X}$ if the formula $\Delta$ is evaluated to be true after substituting the variables by their assignments. For example, given an SMT formula encoding a ReLU, $( (W · 1) > 0 \Rightarrow Z = W · 1) \land (W · 1 ≤ 0 \Rightarrow Z = 0)$, the assignment $(W = 3, Z = 1)$ satisfies the formula since both $(W · 1) > 0 \Rightarrow Z = W · 1)$ and $(W · 1 ≤ 0 \Rightarrow Z = 0)$ are evaluated to be true in this conjunction. Another satisfying assignment would be $(W = -1, Z = 0)$. While an assignment $(W = 3, Z = -1)$ does not satisfy $\Delta$ since $(W · 1) > 0 \Rightarrow Z = W · 1)$ is evaluated to be false, even though $(W · 1 ≤ 0 \Rightarrow Z = 0)$ is evaluated to be true, due to the conjunction of both. We’ll include the definitions and intuitive explanations of the satisfaction of a logical formula in the camera-ready version for readability.
>
> [Q2] One message we would like to deliver in this work is that a closed-form inference, even formed by low-order polynomials, is able to deliver accurate approximation, as shown in Figure 1. To form an approximation to the Gaussian distribution, the Taylor series is not applicable since mathematically it is for approximating the function locally near a certain point instead of approximating the whole distribution. The approximation of Gaussian using a triangular distribution in our work is common in numerical analysis and has been adopted in various applications [3,4].
>
> [Q3] We agree that it will be interesting future work to study the combination of CIBER and temperature scaling.

---

> > ### Comment · Reviewer_cv4P · 2023-08-13
> > **Response to Reviewers**
> >
> > I thank the authors for the response. Some things are more clear now such as the ReLU satisfiability and the runtime complexity. I wasn't convinced that indeed this method is prefered over baseline methods, at least in terms of the empirical results presented in the paper. This is fine though as I mainly judge this method by its novelty and not the empirical evaluation.
> >
> > One thing I do not agree with the authors about is the justification brought here in the rebuttal for using uniform approximation to the posterior. While I do understand the computational constraints forcing the authors to make that choice, I do not think that it holds in terms of a formal justification. Under discrete sampling, the uniform weight makes sense as we assume that the samples reflect high density regions according to the posterior (or its approximation). This assumption does not hold in the case of a continuous uniform distribution. For instance, if the posterior is Gaussian and if we sample it regularly enough, then the predictive distribution will be approximately Gaussian as well. However, this will not be the case with a continuous uniform posterior distribution.
> >
> > Overall, considering the authors response and the reviewers comments I decided to keep my original score.

---

> > > ### Author Response · Authors · 2023-08-14
> > > **Reply**
> > >
> > > We would like to thank you for appreciating the novelty of our work.
> > >
> > > However, the discussion of convergence in the limit entirely misses the point of the paper. You cannot achieve low variance and low bias in Bayesian deep learning. The key point of our paper is that the biased uniform distribution is often more accurate than other methods in the finite sample case. Of course our proposal is biased and that means that in the limit of infinite samples, the uniform approximation is not formally justified. But in the finite sample case it is, as we show. That is the entire point of the paper!

---

> > > > ### Comment · Reviewer_cv4P · 2023-08-19
> > > > **Reply to Authors**
> > > >
> > > > I thank the authors for the clarification. Please include these full explanations in the revised version of the paper. I am not sure that settling on a lower variance approach on the account of a high bias is the path to take, nevertheless its an interesting and original direction in my opinion. Hence, I decided to raise my score to 7.

---

> > > > > ### Author Response · Authors · 2023-08-21
> > > > >
> > > > > We greatly thank you for considering our work to be an interesting and original direction. We'll include these full explanations in the next version of the paper as suggested.

---

### Official Review · Reviewer_PnkR · 2023-07-05

**Soundness:** 3 good
**Presentation:** 3 good
**Contribution:** 2 fair
**Rating:** 6
**Confidence:** 4

**Summary:**

The paper provides a closed-form approximation for the posterior predictive distribution in Bayesian deep learning, in both regression and classification. The paper is overall clear and quite well written. The theory seems reasonable, however a theoretical analysis on the approximation error, depending either on the number of model parameters or on the number of predictive samples, is not provided. Currently, it is not particularly clear to me the sensitivity of the results to the number of samples. As Monte Carlo will eventually win for a large number of samples, I believe it is important to understand up to what point the proposed method is better.

**Strengths:**

The paper is quite well written. It attempts to provide closed-form approximation for the posterior predictive distribution in Bayesian deep learning, which is a meaningful efforts considering that Monte Carlo samples can be expensive. The approximations proposed by the paper seem reasonable. The paper proposes a number of experiments and benchmarks against several methods.

**Weaknesses:**

- The paper lacks of a sensitivity analysis to the number of samples.

- It would be great if the error for some of the approximations could also be expressed analytically. For example, can we say anything about the error of a triangular approximation?

- If we strip the paper of its motivating terminologies (WMI and collapsed samples), seems similar in spirit to partially stochastic Bayesian inference, with a uniform posterior approximation and a triangular approximation of the likelihood to work out a closed-form approximation for the predictive. There does not seem to be a direct connection/comparison with this part of the literature.

**Questions:**


Some more granular comments below.

- I find Figure (2) not very convincing. The authors claim that the predictive uncertainty using 10 samples from CIBER is closer than the one using 10 sample from HMC to the benchmark approximation given by 2k samples from HMC. First of all, the blue prediction from CIBER seems further off than HMC with 10 samples. Consequently, the uncertainty interval is wider. While it is positive that the larger uncertainty is adjusted to to the wider prediction error, to me HMC-10 seems closer to HMC-2000 than CIBER-10. It would also be interesting to see what happens with a few more samples, say 20 or 30, which is still low but often possible. Also, I assume the 10 samples were taken in the asymptotic regime, that is after HMC converged?

- Also in Figure 3, the authors use 5 samples only. As this is very low, it would be interesting to see at what point MC becomes better. E.g. is it 7 samples? 10? 100? 1000? This would give the reader a more clear perspective of when an algorithm should be used instead of the other.

- Line 128: It would be good to expand on the meaning of this definition. The example below clarifies a bit, but, being the terminology non-standard, it is still not immediate to understand what "satisfaction of an SMT formula" means, and the idea underlying the definition of WMI.

- I cannot find in the appendix the piecewise polynomial approximations for classification mentioned at line 213.

- In proposition 7, the authors derive expressions for predictive density and predictive mean. What about other statistics, like variance and entropy? Can these be derived in closed form as well? If so, it would be good to provide an expression.

- To me the definition of collapsed BMA remind a lot partially stochastic neural networks, that is when Bayesian inference is performed on part of the network only. In fact, give a posterior distribution p(w|D), where w=[w_c, w_s], you can always decompose p(w|D)=p(w_c|w_s, D)p(w_s|D). If you approximate p(w_s|D) with an empirical distribution at a bunch of samples, there you have collapsed BMA. In partially stochastic neural networks, usually p(w_s|D) is just a single Dirac delta rather than an ensemble. In this paper, instead, w_c are taken along the trajectory of SGD, similarly as in SWAG. Perhaps it would be worth to explain the connection.

- In the conditional posterior approximation, how are the lower and upper limits of the uniform distribution defined?

- The paper mentions [Kristiadi et al. 2022] a few times. In these paper, the authors also provide a closed-form approximation for the predictive distribution in classification, i.e. the multi-class probit approximation. Since their rationale is similar as for this paper (closed-form approximations in a low-sample regime), it would be good to see a comparison.

**Limitations:**

There does not seem to be much discussion about limitations. The authors could better discuss in which scenarios their approximation is better or worse than other methods.

---

> ### Author Rebuttal · Authors · 2023-08-10
>
> We would like to thank the reviewer for appreciating our work for its clear presentation, solving a meaningful problem, reasonable approximation, and thorough empirical evaluations. In what follows, we will address your concerns, with the references put in the general response due to the character limit.
>
> [Q1] Figure 2 demonstrates that for overconfident ReLU NNs, using HMC with 10 samples does not help the overconfidence issue much: its confidence intervals are irregular and often too narrow, and many predictions have the shaded region much narrower than those in Figure 2c. While in Figure 2b, the shaded region is wide and for the predictions that are more off from the ground truth, the shaded region is even wider meaning that those predictions come with less confidence, showing that using a uniform approximation to posterior helps estimate the uncertainty in a consistent way. Still, we provide a quantitative analysis of the number of samples. Please see our response to [Q2] below.
>
> [Q2] We perform a comparison between CIBER and the MC method for the Bayesian linear regression setting as suggested by the reviewer to see how many samples the MC method needs to match the performance of CIBER with the result presented in Figure A.2 in the rebuttal pdf. The performances of both approaches are evaluated using KL divergence between the estimated posterior distribution and the ground-truth one, averaged over 10 trials. In Figure A.2, the dashed green line shows the performance of CIBER with 50 samples and the blue curve shows the performance of MC with an increasing number of samples. As expected, the MC method yields lower KL divergence as the number of samples increases; however, it takes more than 100 samples to match CIBER, indicating its low sample efficiency and that developing efficient and effective inference algorithms such as CIBER for estimating BMA is a meaningful question.
>
> [Q3] Please see [satisfaction to SMT formulas] in the general response.
>
> [Q4] For classification, a piecewise polynomial approximation with the polynomial degree being three is applied to the sigmoid function which we visually present in Figure A.3 in the rebuttal pdf, where again is obtained by minimizing the L2 distance. We’ll include the explicit expression of the piecewise polynomial approximation in the next version.
>
> [Q5] For variance, we derive the closed-form expression in Proposition B.1 in the rebuttal pdf; this is a direct generalization of our results because variance can be derived from the second moment of the distribution which is an expectation over $y^2$, a polynomial. While for entropy, the existing WMI solver does not allow the log operation and thus there’s no direct result. We do think it is an interesting future research question on what statistics can be exactly/approximately solved by WMI solvers and how such results can motivate the derivation of new algorithms for accurate and reliable inference.
>
> [Q6] The idea of partially stochastic neural networks [4] is to partition the weights into two sets: a deterministic set where the weights are assigned to be MAP estimation and a stochastic set where the weights are stochastic and approximated using existing Bayesian deep learning methods such as Laplace approximation, variational inference or SWAG. This is different from CIBER where the neural network is fully stochastic. CIBER also partitions the weights into two sets: a sampling set where weights are approximated by SWAG and a collapsed set where the weights are approximated using our proposed closed-form inference. A future research direction can be to combine both by applying the closed-form inference to the stochastic set in the partially stochastic neural networks to study if the observations on stochasticity in [4] still hold. We’ll this discussion and the related work on the partially stochastic neural networks including [4] in the camera-ready version.
>
> [Q7] The lower and upper limits for each weight are defined by the minimum and the maximum of the weight samples drawn from the SGD trajectories respectively.
>
> [Q8] The closed-form approximation in [2] is motivated by a binary probit approximation when the logit conforms to a Gaussian distribution, that is, the posterior predictive is a Gaussian, which is exactly the same setting we have for the synthetic classification in order to provide a quantitative analysis of the approximation when the BMA ground truth is available. In this setting, the probit approximation is almost exact. Still, such approximation is specific to classification while our proposed approximation using a reduction to weighted volume computation is applicable to layers with various activations as shown in experiments. Also, it assumes mean-field approximation that ignores correlations induced by activations such as ReLU while the ReLU can be naturally encoded in the weighted model integration framework.
>
> [piecewise-polynomial approximation]
> The piecewise polynomial approximation enjoys theoretical guarantees on approximation error by the Stone-Weier theorem [4], stating that any continuous functions can be uniformly approximated by polynomials up to arbitrary precision, which holds for high dimensional space. That is, functions $f$ including multivariate Gaussian can all be effectively approximated by piecewise polynomials, with theoretical guarantees that for arbitrary $\epsilon$, there exists a piecewise polynomial $p$ such that $\parallel f - p \parallel < \epsilon$ holds, where $\parallel \cdot \parallel$ denotes the supremum norm. In our work, we choose to approximate Gaussian using the triangular distributions which is common in numerical analysis and has been adopted in various applications [5,6].

---

> > ### Comment · Reviewer_PnkR · 2023-08-16
> >
> > I thank the reviewers for the additional experiments and the explanations. As a result, I have increased my score to 6.

---

### Official Review · Reviewer_e8p8 · 2023-07-06

**Soundness:** 3 good
**Presentation:** 4 excellent
**Contribution:** 3 good
**Rating:** 6
**Confidence:** 3

**Summary:**

The paper proposes to use techniques from weighted volume computation to deterministically marginalize over (a subset of) the weights in a BNN rather than sampling them when computing the predictive posterior. The experiments find the method to perform competitively with some standard baselines from the literature on UCI and CIFAR benchmarks

I think this is a nice methodological contribution, although I wish the experiments reported the computational cost and explored the impact of only being able to select a subset of the weights for volume computation (i.e. more detailed analysis of the limitations). So overall I would lean towards acceptance.

**Strengths:**

* Using volume computation for BNNs is new as far as I am aware, and I find work that connects different sub-fields in the literature generally quite valuable.
* The paper is very well written, I found the low-dimensional running example used throughout the paper very effective.
* The method seems to perform well in the experiments.

**Weaknesses:**

* The paper mentions that applying volume computation over all weights would not be feasible, but does not report how expensive it is on the subset that it select.
* Similarly, I would have liked to see some analysis of the compute time-performance tradeoff as one varies the number of weights that are used in volume computation rather than sampled.

**Questions:**

I’d mainly like to see some discussion on the points mentioned in the weaknesses.


**Limitations:**

The computational cost is mentioned but not further substantiated by runtime figures and comparisons.

---

> ### Author Rebuttal · Authors · 2023-08-10
>
> We deeply thank the reviewer for appreciating our work for its novelty, clear presentation, and thorough empirical evaluations. We truly appreciate that you find the connection between sub-fields proposed in our work to be quite valuable. In what follows, we will address your concerns.
>
> [runtime complexity]
>
> We provide the runtime of CIBER on dataset CIFAR-100 dataset in the classification experiment with the three neural network models being VGG-16, PreResNet-164, and WideResNet28x10 respectively as below, to showcase the empirical computational complexity of CIBER. We compare our runtime with SWAG which is known to be a simple and scalable Bayesian deep learning approach summarized in the table below. The runtime of CIBER consists of two main processes: 1) training time, which is exactly the same as the runtime of SWAG, and 2) WMI solving time. The table shows that CIBER is almost as efficient as SWAG and that WMI solving brings little computational overhead due to the efficiency of the WMI solvers.
>
> | MODEL           |   VGG-16   | PreResNet-164 | WideResNet28x10 |
> |-----------------|:----------:|:-------------:|:---------------:|
> | Training / SWAG | 2h3m59s    | 6h3m18s       | 29h39m28s       |
> | WMI Solving     | 11m43.134s | 11m20.638s    | 19m1.076s       |
>
> [runtime-performance tradeoff]
>
> We perform analysis on the runtime-performance tradeoff as suggested by the reviewer, with results presented as Figure A.4 in the rebuttal pdf. The analysis is carried out in the Bayesian linear regression setting where the ground-truth posterior predictive distribution is available. The performance of CIBER is evaluated using the KL divergence between the estimated posterior predictive distribution and the ground truth. As we increase the size of the collapsed set, the BMA integrals are defined over a higher dimension, leading to longer WMI solving time; in the meanwhile, the accuracy brought by the closed-form approximate inference over the collapsed set leads to lower KL divergence. We'll include this analysis in the camera-ready version.

---

> > ### Comment · Reviewer_e8p8 · 2023-08-15
> >
> > Thank you for providing runtimes and figure A.4. I think a similar figure for some NN experiment(s) with e.g. NLL as the performance measure could be helpful for the camera-ready paper (I understand that this may not have been doable over the rebuttal period). I would also suggest adding a secondary x-axis to the figure indicating the number of weights considered for collapsed inference.
> >
> > Overall and in light of the other reviews I remain with my score.

---

### Official Review · Reviewer_eHBs · 2023-07-06

**Soundness:** 3 good
**Presentation:** 2 fair
**Contribution:** 3 good
**Rating:** 4
**Confidence:** 3

**Summary:**

The authors propose to tackle the intractable problem of Bayesian model averaging (BMA) in Bayesian neural networks by re-formulating it as “collapsed BMA”, where a small number of “collapsed” samples from a subset of the parameter space (e.g., the last layer) is equipped with a posterior conditioned on the remainder. The key contribution is to cast the marginalization over this conditional posterior as a weighted volume computation (WVC) problem, computing integrals over feasible regions of arithmetic constraints that are to be inferred from binary ReLU activation patterns. As such enumeration of constraints is computationally infeasible, a weighted model integration (WMI) framework is used to solve BMA in a multi-step procedure of encoding the involved distributions and calling existing WMI solvers. The proposed algorithm competes favorably against a number of baselines.

**Strengths:**

* [S1] **Significance**. The authors address an important problem with a focus on practical application.
* [S2] **Originality**. The proposed solution seems novel and creative.
* [S3] **Structure**. Structural elements like enumerated steps and questions are helpful to follow the argumentation.
* [S4] **Intuition**. The authors make an effort to build intuition with several examples and illustrations.


**Weaknesses:**

* [W1] **Storyline**. While the authors clearly try to establish a coherent structure with well-motivated steps, the many components make it hard to follow and not lose sight of the key idea.
   * The main topic, collapsed inference, is not addressed until page 5 – not being familiar with WVC/WMI, I felt somewhat lost in notation and descriptions of approximate procedures until the concept is even formally introduced.
   * I would recommend to have Def. 6 early on, and to map the 4-step procedure in Section 3 directly to the collapsed BMA problem.
   * I It would be helpful to provide more intuition around all the notation around WVC/WM – to readers mainly familiar with Bayesian inference and Bayesian deep learning, these concepts might appear quite foreign.
   * I Also, if space constraints allow it, I would move the central CIBER algorithm into the main paper.
 * [W2] **Simplifications**. The concept seems to rely on a number of simplifying assumptions that I feel are not properly defended.
   * I Encoding the posterior predictive distribution with polynomial densities
   * I Choosing the conditional posterior, a key object in your proposal, to be uniform with an explanation that reads like “better than not doing it at all”
* [W3] **Evidence**. The empirical results do not quite support the claims made.
   * I Some of the figures/examples do not seem convincing to me, see Q2–Q4.
   * I Your claim of “set[ting] a new state of the art” is not thoroughly funded in my opinion.
     * I cannot find details on how the architecture was chosen for the large datasets.
     * Baselines: In the transfer learning task, the comparison seems limited (SWA and SWAG are quite similar, and I would not consider SGD a “strong” baseline). Also, I miss Laplace approximation and especially deep ensembles as popular inference schemes that arguably operate in the few-sample setting as well.
     * The methods are evaluated on small-scale UCI tasks, and the results are not entirely conclusive.
   * I miss an ablation w.r.t. the number of collapsed samples, which should be an important hyperparameter of your method.
* [W4] **No code**. With a novel method of computing BMAs, it would be helpful to have a look at the code. Also, it would allow to inspect implementation details of the experiments.


**Questions:**

* RE [W2]
  * [Q1] Can you elaborate on the appropriateness for the approximation with polynomial densities, especially for (a) the Gaussian case in high dimensions, and (b) the classification case?
  * [Q2] Have you explored any alternatives to the uniform distribution for the conditional posterior? The empirical evidence provided in Fig. 2 is also not clear to me: (a) How can HMC and CIBER use the “same 10 samples” when HMC recovers the true posterior after convergence, and CIBER samples from the SGD trajectory? (b) I fail to see how Fig. 2b is closer to 2c than 2a.
* RE [W3]
  * [Q3] Fig. 3: Did you repeat the sampling process? If not, the superiority of CIBER established from 5 samples might be a lucky shot.
  * [Q4] Classification example (l. 240): Kindly elaborate on this, I do not understand the origin of the presented numbers.
* [Q5] l. 37: How does collapsed sampling limit to a subset of *variables*? From your descriptions, this should be a subset of *weights*.
* [Q6] l. 99: Should one of the inequality signs be a strict “greater” or “lesser”?
* [Q7] l. 161: CIBER claims to resolve the issue of WMI-encoding non-ReLU networks. How is this achieved? Proposition 7 still contains * $\Delta_{ReLU}$.
* [Q8] Eq. 2: The sum is over tuples $(w_s, q)$ but $w_s$ never appears in the summand and $q$ is the same for all $w_s$. Can you make more explicit how the running index enters the sum? On a related note, I do not understand the definition of $W_s$ (l. 183) because I don’t quite see how $w_s$ and $w$ are connected.
* [Q9] l. 283: Could you elaborate on what you mean by “greater variance indicates that the weight is prone to have greater uncertainty and thus one might want to perform more accurate inference over it”?
* [Q10] l. 314: Should the number of collapsed samples not depend on network architecture rather than number of classes in the final layer?

—

Minor remarks

* l. 31: I would not agree that “existing methods mainly focus on MCMC” in BNN; approximate schemes like deep ensembles are quite popular.
* l. 39: The claim that “collapsed samplers are effective at variance reduction in graphical models” seems irrelevant in the context of your proposal.
* l. 62: “it can be risky to base inference on a single neural network model” sounds fairly hand-wavy.
* l. 79 (example 2): The notation is mixing up bold and non-bold w.
* l. 148: How is that the “uncertainty” of the prediction?
* l. 183: The definition of $W_s$ seems overly complicated, why not just say “given a set of parameter samples $W_s$”?
* l. 265: What is DPLL?
* l. 272: I would recommend removing the “Boston” dataset due to its racism issues.
* l. 300: There seems to be an “and” missing before “four”.
* l. 302: “root mean-squared error”, not “rooted-mean-squared-errors”.
* Some references could be added in
  * l. 35/36 (cutset / Rao-Blackwellised samplers)
  * l. 51 (HMC)
  * l. 110/111 (“various” solvers; single reference)

**Limitations:**

I miss a discussion of the limitations, especially since there are numerous approximations and heuristics involved. Societal impacts (or rather, the absence of such) are indeed mentioned.

---

> ### Author Rebuttal · Authors · 2023-08-10
>
> We greatly thank the reviewer for appreciating our work for its significance, originality, novelty, and structures. We will address your concerns below, with references in the general response.
>
> [Questions on W2, W3]
>
> [Q1] The appropriateness of piecewise polynomial approximation is justified by the Stone-Weier theorem [1], stating that any continuous functions can be uniformly approximated by polynomials up to arbitrary precision, which holds for high dimensional space. That is, functions $f$ including multivariate Gaussian and softmax can all be effectively approximated by piecewise polynomials, with theoretical guarantees that for arbitrary $\epsilon$, there exists a piecewise polynomial $p$ such that $\parallel f - p \parallel < \epsilon$ holds, where $\parallel \cdot \parallel$ denotes the supremum norm.
>
> [Q2] We agree that to explore what are the alternative approximations to the posterior is interesting for future work; in our work, it is chosen to be uniform, and we provide a further justification for this choice in the general response.  In Figure 2, 10 samples are drawn from a posterior distribution where HMC uses them for inference while CIBER first applies a uniform approximation to them before the closed-form inference. This figure demonstrates that for overconfident ReLU NNs, using HMC with 10 samples does not help the overconfidence issue much: its confidence intervals are irregular and often too narrow, and many predictions have the shaded region much narrower than those in Figure 2c. While in Figure 2b, the shaded region is wide and for the predictions that are more off from the ground truth, the shaded region is even wider meaning that those predictions come with less confidence, showing that using a uniform approximation to posterior helps estimate the uncertainty in a consistent way.
>
> [Q3] We repeat the sampling for the synthetic Bayesian linear regression experiment for 10 trials as suggested by the reviewer. The resulting comparison of the predictive distributions is presented in Figure A.1 in the rebuttal pdf, where the estimation by CIBER is much closer to the ground truth posterior predictive distribution than the Monte Carlo method. Further, the averaged KL divergence between the ground truth and CIBER is 0.069 while the one for MC estimation is 0.130, again indicating that CIBER yields a better BMA approximation in the few-sample setting.
>
> [Q4] Starting from Line 240, we aim to provide a comparison between CIBER and the Monte Carlo method in the classification setting. A previous work [2] proposes that such comparison on BMA approximation can be achieved by comparing how close an estimation is to the ground-truth integral as defined in Line 241. We follow the setting in [2] and the results show that the integral estimation $I_c$ by CIBER is closer to the ground-truth integral $I$ than the MC estimation $I_{\mathit{MC}}$.
>
> [Q5] We mention “limiting sampling to a subset of variables” only in Line 37 in the context of the collapsed sampler in the probabilistic graphical model literature and in the remaining of the paper, it is mentioned as the sampling of parameters in the context of Bayesian deep learning.
>
> [Q6] We’ll fix this typo.
>
> [Q7] This issue is resolved by the sampling part in the collapsed samples. Specifically, we propose to use collapsed samples to combine the strengths from two worlds: the scalability and flexibility from sampling and the accuracy from WMI solvers, where flexibility refers to the fact that given a neural network that might contain layers not amenable to WMI encoding, we can sample a subset of network weights including those in such layers, and further condition the network on the sample, to result in a sub-network that is amenable to the WMI encoding.
>
> [Q8] As mentioned in Definition 6, $(\mathbf{W}_s, \mathbf{W}_c)$ is a partition of parameters $\mathbf{W}$, meaning that $\mathbf{W}_s \cup \mathbf{W}_c = \mathbf{W}$ and $\mathbf{W}_s \cap \mathbf{W}_c = \emptyset$. With lower case denoting the assignment, it holds that $\mathbf{w}_s \cup \mathbf{w}_c = \mathbf{w}$ and $\mathbf{w}_s \cap \mathbf{w}_c = \emptyset$, that is, $\mathbf{w}_s$ is implicit in Equation 2. We provide an expanded version of this equation in Section B in the rebuttal pdf to clear this confusion.
>
> [Q9] How to choose the collapsed parameter set is an open question in the design of collapsed samplers. One heuristic we provide is to choose the weight parameters whose samples from the SGD trajectory have high variance. This is because the lower the variance a weight has in its samples, the closer the weight is to a deterministic variable, and there is no need to maintain a distribution over deterministic weights. Thus, we choose the weights with high variance in their samples instead since they are further from being deterministic than the others.
>
> [Q10] The numbers in Line 314 are the size of the collapsed parameter sets; for the number of collapsed samples, it is independent of the network architecture and the number of classes, and we keep it the same as the number of samples that the baseline SWAG [3] for a fair comparison.
>
> [On W1]
>
> We agree that the suggested change in the storyline can help readers have a full picture of the collapsed inference scheme before introducing the connection between BMA and WVC, which is easy to make. We would also like to point out that, which part is easier or more important, the collapsed inference scheme vs. the reduction from BMA to WVC, depends on the reader's background and that other readers might prefer the current presentation.
>
> [On W4]
>
> We have provided our code using an anonymous link and have sent the link to AC.
>
> [fianl remark]
>
> We hope that our responses address your concerns and we kindly hope that you would consider raising the score accordingly. We feel that the foundational contribution of this paper has arguably not been fully appreciated, at least not incorporated into the scores, and should be given more weight.

---

> > ### Comment · Reviewer_eHBs · 2023-08-17
> > **Response to rebuttal**
> >
> > Thank you very much for addressing my concerns. Please find below the replies to points that remain open from my perspective.
> >
> > [Q1] My issue was in particular with using a triangular distribution, i.e., a very small number of polynomials, to approximate a Gaussian distribution, a concern also raised by reviewers PnkR and cv4P. I remain skeptical as to how well this approximation will work, especially in higher dimensions. As mentioned several times in the discussion, such approximations should be analyzed quantitatively and discussed in the limitations.
> >
> > [Q2] Thank you for the clarification. However, I don’t quite follow your argument here – if computing an expectation w.r.t. an arbitrary posterior density were equivalent to that w.r.t. a uniform weight distribution, we would hardly face the problem of Bayesian inference at all. Regarding Fig. 2, I see what you mean, but I still disagree with CIBER being “closer” to the 2k-HMC. 10-HMC might indeed be somewhat overconfident but it is not clear to me that CIBER’s being much less confident is more justified – in particular,  the wider confidence interval of CIBER should be considered in the light of its poorer mean approximation.
> >
> > [Q4] Thank you for the clarification. I would strongly recommend to include all the computations (as well as training details, as mentioned by reviewer cv4P) in the supplementary material or provided code.
> >
> > [Q7] I would recommend to make this point, which is certainly quite relevant, more explicit.
> >
> > [Q10] My apologies, I was indeed referring to the size of the collapsed sets, not the number of samples. A follow-up question here: Isn’t using 10 (100) weights in the last layer for CIFAR10 (CIFAR100) simply equivalent to using all (rather than the most variable) weights in the last linear layer?
> >
> > [W1] That is quite true. I was speaking from the perspective of someone mainly interested in Bayesian inference, with WVC only a means to an end, which might apply for others in your target audience.

---

> > > ### Author Response · Authors · 2023-08-18
> > >
> > > We're more than happy to provide further clarifications.
> > >
> > > [Q1] We want to repeat the fact that we know this is a highly biased approximation, intentionally. The main point of our paper is that we choose to have bias over variance and that this works well given the collapsed inference that is then possible. Experiments show that the bias is acceptable as we get strong empirical performance.
> > >
> > > [Q2] We never mean they are equivalent; in most practical Bayesian deep learning settings, there are no explicit posteriors, and thus approximations are required. The key point of our paper is that the uniform distribution as approximations to posteriors is often more accurate than other methods in the finite sample case, as shown in the regression and classification experiments.
> > >
> > > [Q4, Q7] We have provided the code. We'll further augment more training details and the suggested changes in the next version.
> > >
> > > [Q10] The number of total weights at the last layer is n*m, where n is the number of its inputs and m is the number of classes. Thus the weights we choose are subsets of the last-layer weights.
> > >
> > > We hope these convince the reviewer to change their recommendation towards an accept, and are happy to answer any more questions.

---

> > > > ### Comment · Reviewer_eHBs · 2023-08-18
> > > >
> > > > [Q1] Thank you for the clarification; perhaps it would be advisable to state this in more explicit terms. Incidentally, the manuscript doesn't seem to mention the word "bias".
> > > >
> > > > [Q2] I might have misunderstood this point, yet you write in your general response "estimates the posterior predictive distribution by aggregating over a finite number of samples (...), which is equivalent to an expectation (...) under a discrete uniform distribution". Please note also that I'm not condemning such approximations but I do think they should be discussed openly as limitations.
> > > >
> > > > In the light of the discussion and proposed changes I'm raising my score to 4.

---

> > > > > ### Author Response · Authors · 2023-08-21
> > > > >
> > > > > Thanks for raising the score. We'll include these discussions in the next version of the paper as suggested.

---

### Author Rebuttal · Authors · 2023-08-10

We would like to thank all the reviewers for their insightful comments and valuable suggestions, and for deeming our work to be well written (all reviewers) with helpful intuitions (eHBs, cv4P) while presenting a novel and creative view (eHBs,cv4P,e8p8) to an important problem (eHBs,PnkR) that builds a valuable connection between sub-fields (e8p8).

[rebuttal pdf / code]
We’ve attached a rebuttal pdf for the figures. Additionally, we provide our code for all experiments using an anonymous link and have sent the link to AC.

[uniform approximation to posterior]
We would like to provide a further justification for the uniform approximation to posterior using a comparison with the Monte Carlo method as follows. Given samples from a posterior $\mathbf{w_i} \sim p(\mathbf{w}\mid \mathcal{D})$ , a Monte Carlo sampling procedure estimates the posterior predictive distribution by aggregating over a finite number of samples, i.e., $p(y \mid x, \mathcal{D}) = \frac{1}{N} \sum_{i = 1}^n p(y \mid \mathbf{w_i}, x)$, which is equivalent to an expectation of the predictive $p(y \mid \mathbf{w}, x)$ under a discrete uniform distribution $\mathcal{U}(\mathbf{w_1}, \cdots, \mathbf{w_n})$, that is, $p(y \mid x, \mathcal{D}) = E_{\mathbf{w} \sim \mathcal{U}} [p(y \mid \mathbf{w}, x)]$. In CIBER, instead of aggregating over a finite number of weight samples using a discrete uniform distribution, we consider a continuous uniform distribution $\mathcal{U}^\prime[ \mathbf{w_{\mathit{min}}}, \mathbf{w_{\mathit{max}}}]$ where the weight limits are estimated by the samples, to estimate the predictive posterior using a continuous ensemble, i.e., $p(y \mid x, \mathcal{D}) = E_{\mathbf{w} \sim \mathcal{U}^\prime} [p(y \mid \mathbf{w}, x)]$, which can be solved exactly using our propose reduction to WMI problems.

[satisfaction to SMT formulas]
An assignment $\mathbf{x}$ satisfies an SMT formula $\Delta$ defined over variables $\mathbf{X}$ if the formula $\Delta$ is evaluated to be true after substituting the variables by their assignments. For example, given an SMT formula $X_1 + X_2 > 1$, the assignment $(x_1 = 3, x_2 = -1)$ satisfies the formula since $3 + (-1) > 1$ is evaluated to be true, while the the assignment $(x_1 = 0, x_2 = -1)$ does not satisfy the formula since $0 + (-1) > 1$ is evaluated to be false. The WMI problem intuitively defines a conjunction of weighted polytopes where the polytopes are defined by the SMT formulas and the weights are defined by the per-literal weights, and the task of WMI amounts to compute the weighted volume of these polytopes by integrating all the weighted assignments that satisfy the SMT formulas. We agree that these terminologies, which might be common in the logic community, are not familiar to many readers. We’ll include their definitions and intuitive explanations in the camera-ready version.


[limitations]
In this work, we draw the connection between BMA and WVC which inspires us to provide closed-form approximation to BMA integrals using WMI solvers for accurate inference. Still, the WMI encoding of neural network models is only applicable to layers that can be expressed as SMT formulas. Also, the current WMI solvers are limited to polynomial weights, and thus the reduction to WMI problems is applicable to piecewise polynomial weights. This limitation might be alleviated in the future by the development of new WMI solvers that are flexible in the weight function families. We’ll include this discussion of limitations in the camera-ready version.

[References]

[1] De Branges, Louis. "The Stone-Weierstrass theorem." Proceedings of the American Mathematical Society 10.5 (1959): 822-824.

[2] Kristiadi, Agustinus, Runa Eschenhagen, and Philipp Hennig. "Posterior Refinement Improves Sample Efficiency in Bayesian Neural Networks." Advances in Neural Information Processing Systems.

[3] Maddox, W. J., Izmailov, P., Garipov, T., Vetrov, D. P., and Wilson, A. G. A simple baseline for Bayesian uncertainty in deep learning. Advances in Neural Information Processing Systems, 32:13153–13164, 2019.

[4] Sharma, Mrinank, et al. "Do Bayesian Neural Networks Need To Be Fully Stochastic?." International Conference on Artificial Intelligence and Statistics. PMLR, 2023.

[5] Scherer, William T., Thomas A. Pomroy, and Douglas N. Fuller. "The triangular density to approximate the normal density: decision rules-of-thumb." Reliability Engineering & System Safety 82.3 (2003): 331-341.

[6] Karsh, P. K., T. Mukhopadhyay, and S. Dey. "Stochastic low-velocity impact on functionally graded plates: Probabilistic and non-probabilistic uncertainty quantification." Composites Part B: Engineering 159 (2019): 461-480.

[7] Riquelme, Carlos, George Tucker, and Jasper Snoek. "Deep Bayesian Bandits Showdown: An Empirical Comparison of Bayesian Deep Networks for Thompson Sampling." International Conference on Learning Representations. 2018.

[8] Izmailov, Pavel, et al. "Subspace inference for Bayesian deep learning." Uncertainty in Artificial Intelligence. PMLR, 2020.

---

### Decision · Program_Chairs · 2023-09-21

**Decision:**

Accept (poster)

**Comment:**

In the rebuttal authors satisfactorily replied to open questions and concerns, leading all reviewers to increase there scores, three of which vote for acceptance. The authors should definitely apply the promised changes in the revised version of their paper.